# Whole-genome sequencing analysis of semi-supercentenarians

**Paolo Garagnani[1,2,3†]\*, Julien Marquis[4†‡], Massimo Delledonne[5†], Chiara Pirazzini[6], Elena Marasco[1,7], Katarzyna Malgorzata Kwiatkowska[1], Vincenzo Iannuzzi[3], Maria Giulia Bacalini[6], Armand Valsesia[4], Jerome Carayol[4], Frederic Raymond[4], Alberto Ferrarini[5§], Luciano Xumerle[5], Sebastiano Collino[4], Daniela Mari[8], Beatrice Arosio[8,9], Martina Casati[8], Evelyn Ferri[8], Daniela Monti[10], Benedetta Nacmias[11,12], Sandro Sorbi[11,12], Donata Luiselli[13], Davide Pettener[14], Gastone Castellani[1], Claudia Sala[15], Giuseppe Passarino[16], Francesco De Rango[16], Patrizia D'Aquila[16], Luca Bertamini[1,17], Nicola Martinelli[17], Domenico Girelli[17], Oliviero Olivieri[17], Cristina Giuliani[14,18†], Patrick Descombes[4†], Claudio Franceschi[1,6,19†]**

[1]Department of Experimental, Diagnostic, and Specialty Medicine (DIMES), University of Bologna, Bologna, Italy; [2]Clinical Chemistry, Department of Laboratory Medicine, Karolinska Institutet at Huddinge University Hospital, Stockholm, Sweden; [3]Alma Mater Research Institute on Global Challenges and Climate Change (Alma Climate), University of Bologna, Bologna, Italy; [4]Nestlé Research, Société des Produits Nestlé SA, Lausanne, Switzerland; [5]Functional Genomics Laboratory, Department of Biotechnology, University of Verona, Verona, Italy; [6]IRCCS Istituto delle Scienze Neurologiche di Bologna, Bologna, Italy; [7]Applied Biomedical Research Center (CRBA), S. Orsola-Malpighi Polyclinic, Bologna, Italy; [8]Fondazione Ca' Granda, IRCCS Ospedale Maggiore Policlinico, Milan, Italy; [9]Geriatric Unit, Department of Clinical Sciences and Community Health, University of Milan, Milan, Italy; [10]Department of Experimental and Clinical Biomedical Sciences "Mario Serio", University of Florence, Florence, Italy; [11]Department of Neuroscience, Psychology, Drug Research and Child Health, University of Florence, Florence, Italy; [12]IRCCS Fondazione Don Carlo Gnocchi, Firenze, Italy; [13]Department for the Cultural Heritage (DBC), University of Bologna, Ravenna, Italy; [14]Department of Biological, Geological, and Environmental Sciences (BiGeA), Laboratory of Molecular Anthropology and Centre for Genome Biology, University of Bologna, Bologna, Italy; [15]Department of Physics and Astronomy, University of Bologna, Bologna, Italy; [16]Department of Biology, Ecology and Earth Sciences, University of Calabria, Rende, Italy; [17]Department of Medicine, Unit of Internal Medicine, University of Verona, Verona, Italy; [18]School of Anthropology and Museum Ethnography, University of Oxford, Oxford, United Kingdom; [19]Department of Applied Mathematics and Laboratory of Systems Biology of Aging, Lobachevsky University, Nizhny Novgorod, Russian Federation

**\*For correspondence:** paolo.garagnani2@unibo.it

[†]These authors contributed equally to this work

**Present address:** [‡]Lausanne Genomic Technologies Facility, University of Lausanne, Lausanne, Switzerland; [§]Menarini Silicon Biosystems SpA, Castel Maggiore, Italy

**Abstract** Extreme longevity is the paradigm of healthy aging as individuals who reached the extreme decades of human life avoided or largely postponed all major age-related diseases. In this study, we sequenced at high coverage (90X) the whole genome of 81 semi-supercentenarians and supercentenarians [105+/110+] (mean age: 106.6 ± 1.6) and of 36 healthy unrelated geographically matched controls (mean age 68.0 ± 5.9) recruited in Italy. The results showed that 105+/110+ are

characterized by a peculiar genetic background associated with efficient DNA repair mechanisms, as evidenced by both germline data (common and rare variants) and somatic mutations patterns (lower mutation load if compared to younger healthy controls). Results were replicated in a second independent cohort of 333 Italian centenarians and 358 geographically matched controls. The genetics of 105+/110+ identified DNA repair and clonal haematopoiesis as crucial players for healthy aging and for the protection from cardiovascular events.

## Introduction

The study of healthy aging is of increasing importance since the phenomenon of human aging is inevitably linked to cumulative burden of age-associated diseases – such as cardiovascular disease (CVDs), stroke, type 2 diabetes, hypertension, different type of cancer, or dementia (*Christensen et al., 2009*; *Franceschi and Bonafè, 2003*). The geroscience perspective suggested to consider ageing as the major common risk factor for several chronic diseases and conditions (*Kennedy et al., 2014*). However few genetics studies followed this theory to elucidate the common mechanisms between aging and age-related diseases.

The geroscience approach may be applied to many diseases and many experimental designs. Here, we decided: (i) to select an informative model of extreme longevity; (ii) to use a whole genome sequencing at high coverage approach; (iii) to analyze the link with the genetic determinants of CVDs.

The study of human extreme longevity constitutes a model useful to assess the impact of genetic variability on this trait according to the following considerations. First, *Sebastiani et al., 2016* showed that, considering individuals surviving to age 105 years, the relative risk of sibling surviving to 105 years is 35 times the chance of living to age 105 of the control population. These data suggest a more potent genetic contributions if samples are recruited in the last percentile of survival in accord with *Tan et al., 2008* who reported that the power to detect association with longevity is greater for centenarians versus nonagenarians samples of the same birth cohort. Second, despite different definitions and opinion regarding the concept of healthy aging, the clinical and biochemical data on centenarians showed that they can be considered as a paradigm of healthy aging as they avoid or largely postpone all major age-related diseases (*Andersen et al., 2012*). Thus, healthy aging and exceptional longevity (people who live more than 100 years) are deeply related (*Christensen and McGue, 2016*).

Many approaches applied in the last decades to the study of the genetics of human longevity seem to have many limitations, as extensively described (*Sebastiani et al., 2017*). Heterogeneity of the groups – in terms of birth cohort and of population variability – seems to play the most problematic role when different cohorts and datasets are put together in order to increase statistical power. This approach identified genes and pathways important for longevity and healthy aging that are common between human populations, but at the same time misses the context, that is the 'ecological' dimension of healthy aging and longevity (*Giuliani et al., 2017*). In this view, the genetic determinants of longevity are dynamic and historically dependent (*Giuliani et al., 2017*; *Giuliani et al., 2018a*; *Yashin et al., 2015*) and, while the genetic determinants of longevity may be shared by different populations, population-specific genes are expected to play a major role (*De Benedictis and Franceschi, 2006*; *Zeng et al., 2016*).

From *a technological point of view*, the decreasing cost of genotyping arrays has allowed in-depth study of the genetic variability of common variants, using increasingly dense microarrays (>4M SNPs). However, whole genome sequencing (WGS) constitutes a major approach to study genomic variability of each individual (both in coding and noncoding regions). In the study of the genetics of human longevity, there are to date only few examples of WGS. The first studies were published in 2011 by Sebastiani and colleagues who characterized two supercentenarians, in 2014 by Gierman and colleagues *Gierman et al., 2014* who published a study on 13 supercentenarians (110 years or older) and in 2014 considering 44 Ashkenazi Jewish centenarians (*Freudenberg-Hua et al., 2014*). These studies analyzed long-living people without considering a group of controls from the general population, thus reducing the number of potential new information which could be obtained. In 2016 Erickson and colleagues published a WGS paper on a high number of old individuals (N = 511, median age = 84.2 ± 9.3 years) whose health was assessed by self-reported data

('Welldery') and 686 younger controls (median age = 33.3 years) (*Erikson et al., 2016*). However, despite the potential of the technological approach, the relative 'young' age of the elderly, the low number of centenarians and the limitations of the self-reported health status suggest that the possibility to identify the contribution of genetics to human longevity of this study was limited, as argued by *Sebastiani et al., 2017*.

CVDs constitute the first cause of death globally and many studies highlighted the intersection between CVDs and aging as cardiac and vascular aging are considered the major risk factor for CVDs. Many molecular mechanisms have been described as hallmarks of this process such as cellular senescence, genomic instability, chromatin remodeling, macromolecular damage and mitochondrial oxidative stress perturbed proteostasis, vascular and systemic chronic inflammation, among others (*Furman et al., 2019*). An emerging common mechanism between aging and CVD is the accumulation with age of somatic mutations. An age-related expansion of hematopoietic clones characterized by disruptive somatic mutations in few recurrent genes (such as DNMT3A, TET2, ASXL1, PPM1D, TP53), conferring to the mutated cells a selective proliferative advantage, has been described (*Jaiswal et al., 2014*). The expansion of such mutated clones ('clonal hematopoiesis of indeterminate potential', CHIP), has been associated to an acceleration of the atherosclerotic process, an increased risk of haematological malignancies (hazard ratio 11,1), ischemic stroke (hazard ratio 2,6), coronary heart disease (hazard ratio 2,0) and all-cause mortality (*Jaiswal et al., 2014*).

In this study, we generated and analyzed the first WGS data with high coverage (90X) in a cohort of 81 semi-supercentenarians and supercentenarians [105+/110+] (mean age: 106.6 ± 1.6) recruited across the entire Italian peninsula together with a control cohort of 36 healthy geographically matched individuals (Northern, Central, and Southern Italy) (mean age 68.0 ± 5.9). Data recently published (*Giuliani et al., 2018b*) with a second independent cohort of 333 centenarians (>100 years) and 358 geographically matched controls (Northern, Central, and Southern Italy) were used to replicate our results. In order to reduce the heterogeneity of the group we focused on the Italian peninsula as it has been fully characterized in term of genetic structure by different studies (*Sazzini et al., 2016*).

The aim of this study is to identify the genetic determinants of extreme longevity in humans focusing on common and rare variants analysis, 105+/110+ private mutations and somatic mutations, and determining polygenic risk score for cardiovascular diseases, the first cause of mortality in humans.

## Results

### Design of the study and criteria for longevity definition

To study the genetics of longevity, we selected a population of 105+/110+ (N = 81) with mean age of 106.6 years and born in a limited birth cohort range (1903–1909). We used the data produced by WGS in 105+/110+ as discovery data (Cohort 1) to have information about all the variants, and to maximize the probability to identify significant genetic association according to biological models. This design is supported by data showing that the power to detect association with longevity is greater for centenarians versus nonagenarians subjects of the same birth cohort (*Tan et al., 2008*).

The choice of controls is an issue often debated in longevity studies. A number of unrelated samples from the general population matched for geographical origin has been included as control (N = 36). The prevalence of individuals that will become centenarians and semi-supercentenarians in the control group is negligible because of the rarity of the trait. To validate the results obtained in Cohort 1 (discovery phase), we used Cohort 2 (data produced by CoreExomeChip v.1.1 array Illumina 550 k, see details in Materials and methods) that is characterized by a high number of samples (333 centenarians and 358 controls). The design of Cohort 1 (discovery) and Cohort 2 (validation) is driven by the aim of this study, that is to investigate genetic determinants of extreme longevity as a paradigm of healthy aging. A geographical distribution of the samples selected for the discovery and an overview of the study design are reported in *Figure 1A and B* respectively. *Figure 1C* reports PCA plot described in detail in Materials and methods (considering all 117 individuals after quality controls), where individuals from Southern and Northern Italy have been indicated, and we validated that the discovery cohort follows the expected genetic structure (*Raveane et al., 2019*; *Sazzini et al., 2016*).

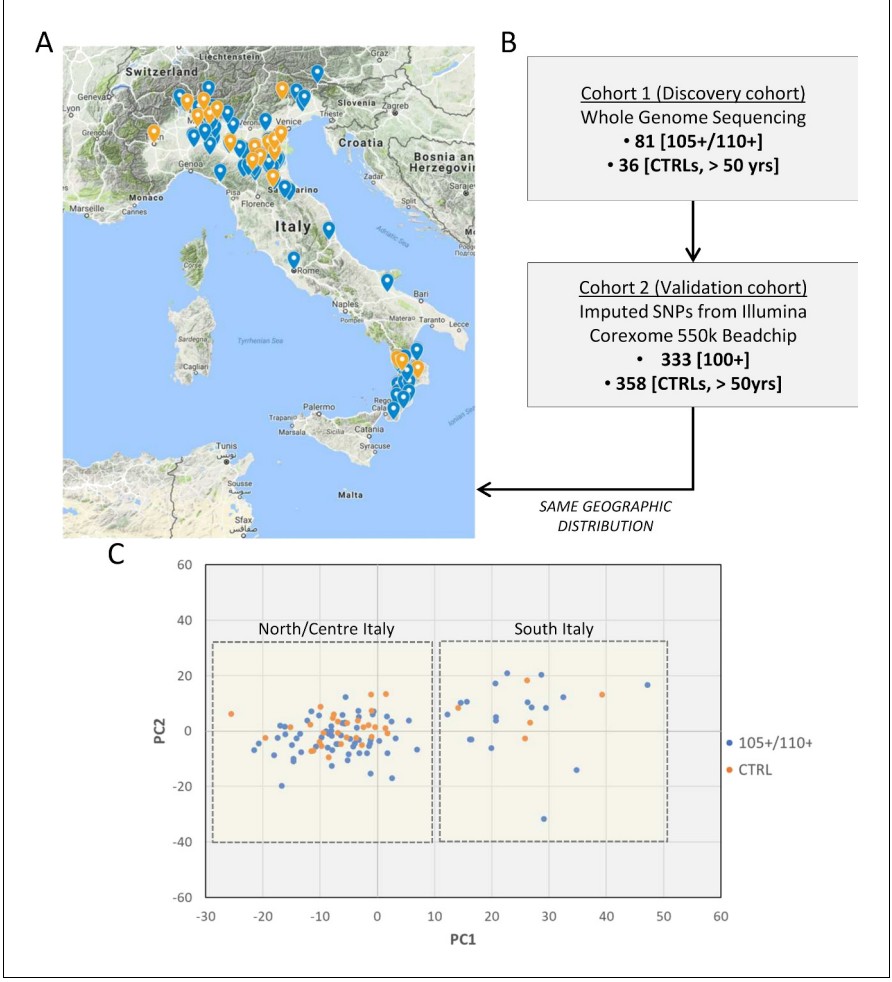

**Figure 1.** Study design. (**A**) 105+/110+ (in blue) and controls (in orange) recruited in the Italian peninsula and analyzed by whole genome sequencing (discovery cohort). (**B**) The study design applied in the present study. (**C**) PCA plot for the discovery cohort (Cohort 1), in red are indicated 105+/110+ and in black the group of controls (CTRL).

## Common variants : single variant analysis

First, we performed a WGS association study adding sex as covariate and considering 5,511,852 common variants (MAF >5%). No association is observed at the genome-wide significance level ($<5*10^{-8}$) (*Figure 2A*), but significant p-values after SLIDE correction (significance at adjusted p-value 10%) are observed for STK17A and COA1 genes. The uniform deflation observed in the QQ-plot (*Figure 2B*) could be due to the small sample size. Genomic inflation factor is 1.02.

Top association signals are located in the same large block of linkage disequilibrium at chromosome seven surrounding STK17A and COA1 genes and are reported in *Table 1* (loci with significance at adjusted p-value 10% for the discovery). Technical validation with different technologies (Sequenom MassARRAY iPLEX) of the identified SNPs was performed considering a subset of 53 individuals from Cohort 1. Positions identified by comparing 105+/110+ and controls with unadjusted p-value$<10^{-4}$ are reported in *Supplementary file 1*. *Supplementary file 1* also includes the previous nominal p-values adjusted for PC1 and PC2.

Given the presence of the well-known Italian genetic structure we decided to perform an association analysis comparing Northern 105+/110+ to Southern 105+/110+subjects. We next merged the identified signals (derived from the comparison 105+/110+ vs CTRL) with the p-values calculated comparing Northern 105+/110+ and Southern 105+/110+ subjects. All the identified signals

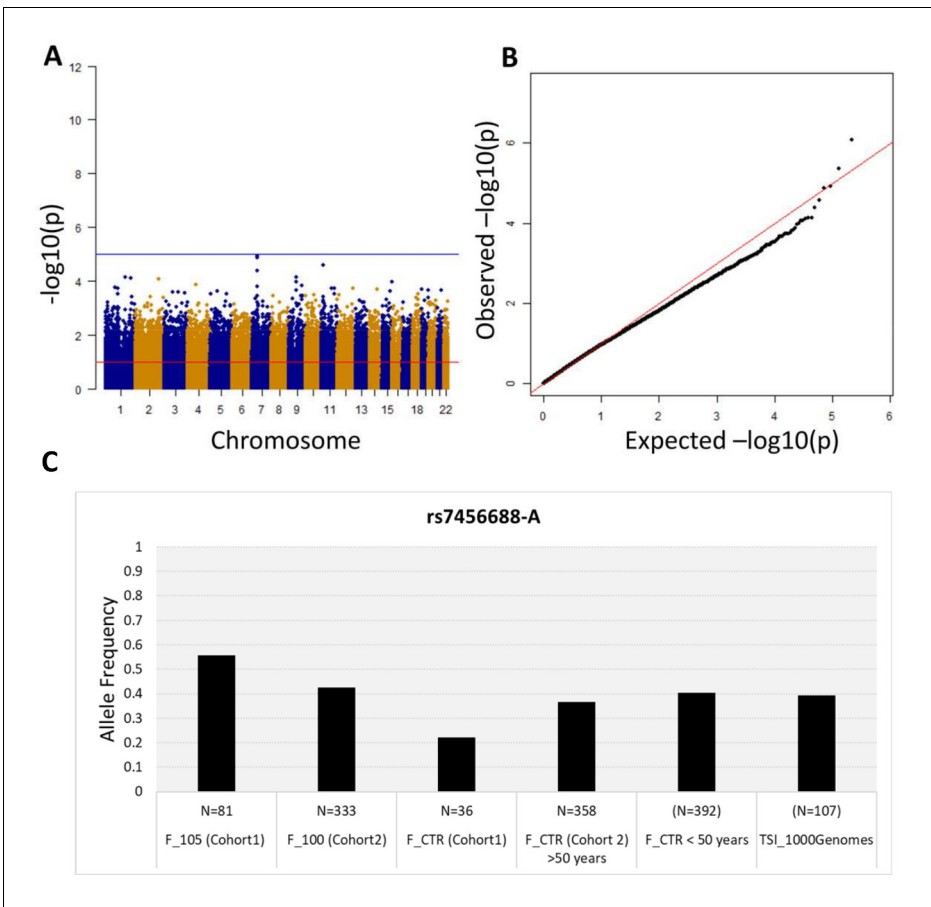

**Figure 2.** Association analysis results considering common variants (MAF >5%). (**A**) Manhattan plot for all the SNPs tested for the association analysis by considering semi-supercentenarians and controls. The x-axis shows SNPs according to their chromosomal positions and y-axis shows the p-values, expressed as –log10(p-value). (**B**) QQ plot of expected –log10(p-values) (x axis) versus observed –log10(p-values) (y axis) (one black point per variant). The genomic inflation factor was estimated to 1.02. (**C**) Allele frequency of rs7456688-A in all the cohorts analyzed. The online version of this article includes the following figure supplement(s) for figure 2:

**Figure supplement 1.** eQTL violin plot for rs623108 (chr7: 43864699) identified in a previous longevity study of the Italian population *Giuliani et al., 2018b* and replicated in the present study.

**Table 1.** Common variants identified in the comparison between 105+/110+ and controls with significance at adjusted p-value 10%. Gene name, chromosome, position (GrCH 37/hg19), rs ID, minor allele (based on whole sample), estimated odds ratio for Cohort 1, lower/upper bound of 95% confidence interval for odds ratio, nominal p-values, adjusted p-values using SLIDE method (window of 100 SNPs an 10,000 permutations) frequency in 105+/110+ s and controls and p-values in Cohort 2 were reported.

| GENE NAME | CHR | BP | dbSNP | A1 | OR | L95 | U95 | P_unadj (Cohort1) | P_adjusted SLIDE (Cohort1) | F_105 (Cohort1) | F_CTRL (Cohort1) | P_unadj (Cohort2) |
|---|---|---|---|---|---|---|---|---|---|---|---|---|
| STK17A | 7 | 43637796 | rs7456688 | A | 5.906 | 2.688 | 12.97 | $9.73*10^{-6}$ | $7.00*10^{-2}$ | 0.556 | 0.222 | 0.021 |
| STK17A | 7 | 43638009 | rs10257700 | C | 5.906 | 2.688 | 12.97 | $9.73*10^{-6}$ | $7.00*10^{-2}$ | 0.556 | 0.222 | 0.029 |
| STK17A | 7 | 43643835 | rs10279856 | G | 5.906 | 2.688 | 12.97 | $9.73*10^{-6}$ | $7.00*10^{-2}$ | 0.556 | 0.222 | 0.021 |
| STK17A, COA1 | 7 | 43650221 | rs69685881 | A | 5.906 | 2.688 | 12.97 | $9.73*10^{-6}$ | $7.00*10^{-2}$ | 0.556 | 0.222 | 0.025 |
| STK17A, COA1 | 7 | 43651047 | rs7805969 | A | 5.906 | 2.688 | 12.97 | $9.73*10^{-6}$ | $7.00*10^{-2}$ | 0.556 | 0.222 | 0.016 |

described in *Supplementary file 1* showed a non-significant p-values in the analysis based on geography. This indicates that the Italian genetic structure does not bias the signals identified.

The five variants (rs7456688, rs10257700, rs10279856, rs69685881, and rs7805969) identified in Cohort 1 were evaluated for association with longevity in the validation Cohort 2. Cohort 2 includes 333 centenarians (mean age: 100.4 ± 1.4) and a group of 358 unrelated healthy individuals (controls; mean age: 60.7 ± 7.2), genotyped on the Illumina 550 k array platform. The analysis of Cohort 2 imputed genotypes (see Materials and methods) returned nominal p-values<0.05 for all the five variants, as reported in *Table 1*. *Figure 2C* reports the allele frequency of rs7456688-A in individuals from Cohort 1 (105+/110+ and CTRL) and Cohort 2 (100+ and CTRL), and of another cohort of healthy controls < 50 years (N = 392, mean age = 39.5 ± 7.2) already described in *Giuliani et al., 2018b*. The Tuscans data (TSI) reported in 1000 Genomes project is also included. The pattern follows a U-shaped with the highest allele frequency observed for 105+/110+, indicating the relevant role of the variant in extreme longevity.

Regional plots (from chr7:43560257 to chr7:43938230) for the most significant regions identified by WGS in Cohort 1 and tested in Cohort 2 (validation cohort) are reported in *Figure 3*. rs2108078, located in chr7: 43861921, was the most significant SNP in this area (p-value=4,2*10$^{-4}$) for the validation cohort.

Next, we performed a gene-based association study by VEGAS2 including common variants (MAF >5%) in Cohort 1. A full list of all the 179 significant genes (p-values<0.01) is reported in *Supplementary file 2*, two of them (APOC3 and PPARGC1A) were already described in the GenAge database (http://genomics.senescence.info/genes/human.html), a database that includes genes identified in studies on human aging. Among the 179 genes identified in Cohort 1 (p-value<0.01, *Supplementary file 2*), eight were also significant in Cohort 2 (p-value<0.05). Fisher method was applied to calculate combined-p-values for STK17A, BLVRA, MYRF, DNAH7, LOC553103, PHF14, SLC22A4, TBRG4 (p-values=2.03*10$^{-9}$, 1.35*10$^{-7}$, 1.83*10$^{-7}$, 0.00049, 0.00146, 0.00205, 0.00154, 0.00146, respectively) and STK17A gene is the gene with the strongest association signal in Cohort 1 (p-value=8.40*10$^{-5}$) and Cohort 2 (p-value=1*10$^{-6}$). LPPR1 was identified in our gene-based analysis of WGS data (VEGAS p-values=7.20*10$^{-5}$) but was not significant in Cohort 2 (VEGAS p-values=0.19).

We then performed a RiVIERA analysis – a tool for variant prioritization – on WGS data for inference of possible causal/regulatory variants considering SNPs located in the above-mentioned window (from chr7:43560257 to chr7:43938230) around the COA1 gene (*Figure 4A*). As reported in *Figure 4A and* SNPs, rs10279856 (in LD with the five previously reported SNPs), rs3779059, rs849166, rs849175 showed a credible score >0 (credible scores 0.261; 0.261; 0.240; 0.236 respectively).

These four SNPs (rs10279856, rs3779059, rs849166, and rs849175) were identified in GTEx as eQTL for STK17A and surrounding genes like COA1 and BLVRA as reported in *Supplementary file 3*. The genotypes the most frequent in centenarians (rs10279856-G reference allele and rs3779059-A, rs849166-A, rs849175-A alternative alleles) are associated to a reduced expression of COA1 gene in adipose (subcutaneous), artery - (aorta and tibial), artery - tibial, esophagus - mucosa, oesophagus - muscularis, nerve - tibial and skin, the same SNPs are associated to an increase in BLVRA expression in whole blood and a decrease of the expression of the same gene in artery (tibial) and oesophagus (mucosa). The same four SNPs are associated to an increase of SKT17A gene expression in heart (atrial and left ventricle), lung, nerve and thyroid.

Longevity is a complex trait for which gene-environment interactions as well as the complex interplay of multiple genes and pathways play a major role (*Zeng et al., 2016*). Gene pathway analysis identified 24 KEGG pathways as significantly enriched (FDR ≤ 0.05) as reported in *Figure 4B* and in *Supplementary file 4*. Axon guidance, calcium signaling, glycine serine and threonine metabolism, long term potentiation, melanogenesis, PPAR signaling and taste transduction are among the most significant pathways identified.

Pathway analysis performed considering GO and BioCarta were reported in *Supplementary file 5* and *Supplementary file 6*. In the top ranking BioCarta pathways enrichment in inflammatory pathways (cytokines and inflammatory response) was observed (FDR value = 0.009).

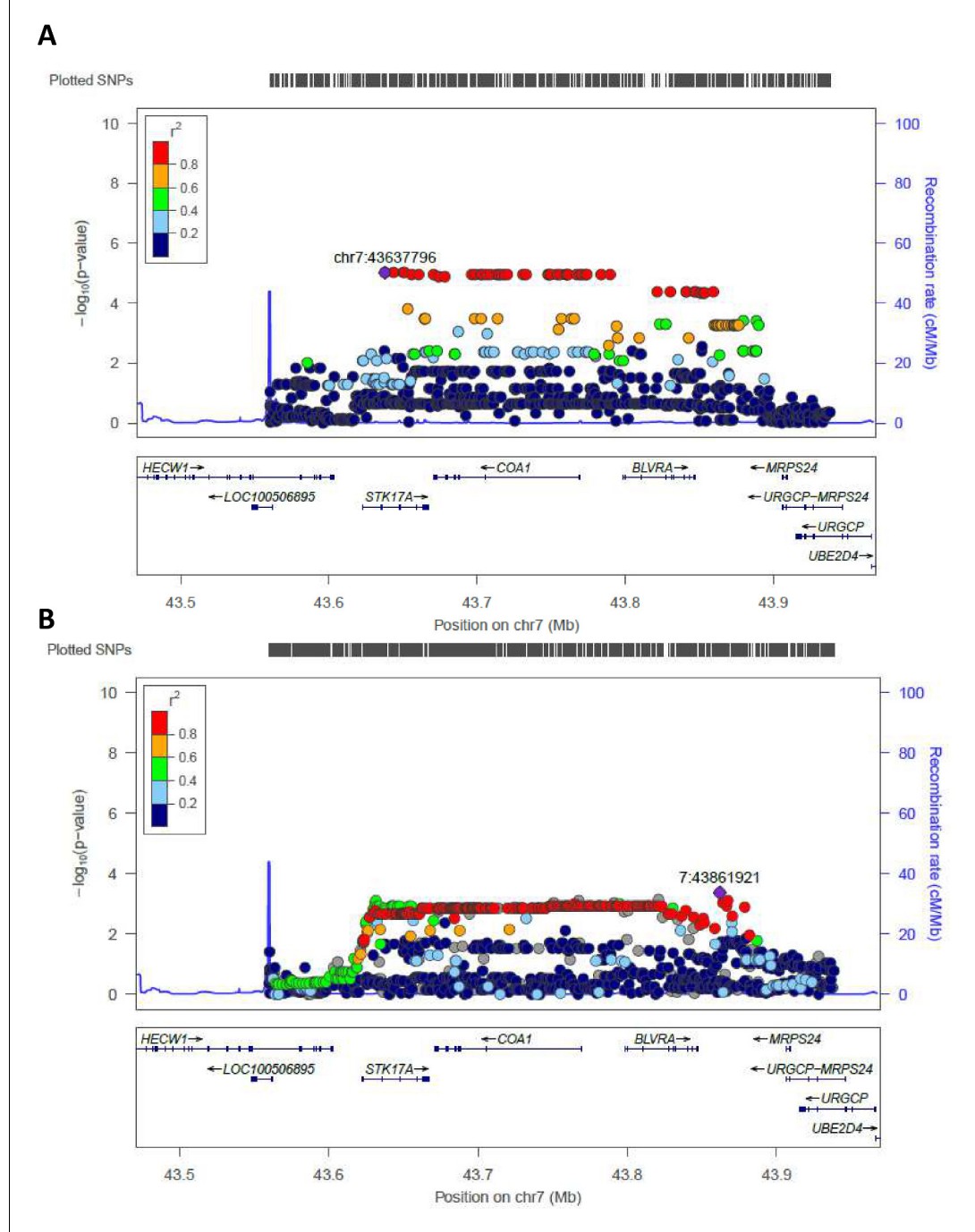

**Figure 3.** Regional association plot made by LocusZoom. Regional plots for the most significant region comparing semi-supercentenarians and controls for the Cohort 1, that is discovery cohort (**A**) and for the validation cohort (**B**). Each point indicates the p-value for one SNP, the x-axis indicates the genomic localization of the variant and the y-axis indicates the -log10(p-value) from the association analysis. The recombination rate is plotted and indicated in the y-axis. Both plots show the same genomic positions, from chr7:43560257 to chr7:43938230 (GRCH37/hg19).

## Common variants: haplotype-based analysis

After genotype phasing, a total of 234 individual haplotypes were used for the haplotype association analysis (162 cases and 72 controls). We analyzed seven suggestive significance areas, emerged from the single-SNP analysis (*Supplementary file 1*), for the association with haplotype clusters. We found a significant association (p-value<5 * $10^{-8}$) for three haplotypes located in COA1 gene

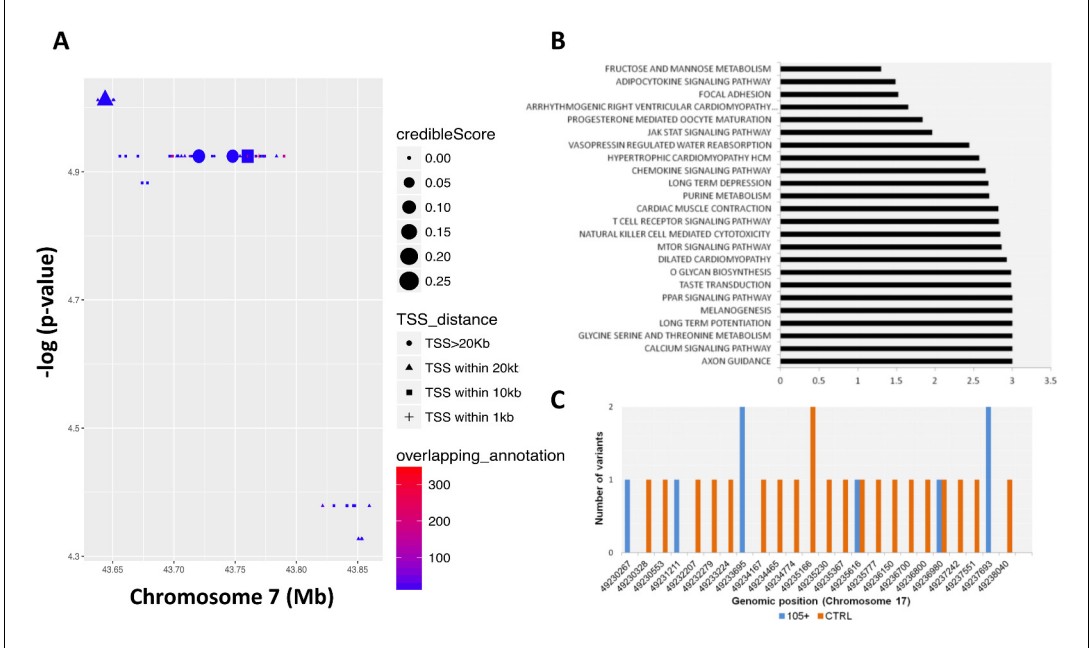

**Figure 4.** Common and rare variants analysis. (**A**) Common variants in COA1 gene and output of the Bayesian model RiVIERA (Risk Variant Inference using Epigenomic Reference Annotations). The SNPs are shown as a function of their position on chromosome 7. The symbol (dot, rectangle, triangle) indicates the distance to the transcription start site (TSS). The size of the symbol reflects the credible score which exhibit an higher probability of regulatory properties. The colour (indicated as 'overlapping_annotation') indicates the total number of epigenomic marks that co-localize with the SNP. (**B**) KEGG Pathways analysis was performed using i-GSEA4GWAS. -log(FDR value) were indicating for each significant pathways (<0.01). (**C**) Number of rare variants in the NME1, NME1-NME2 region. Genomic positions were reported in x-axis while the number of variants for each position is reported in y-axis. The number of rare variants in 105+/110+ is reported in blue and in CTRL in orange.

The online version of this article includes the following figure supplement(s) for figure 4:

**Figure supplement 1.** Number of private mutations for each 105+/110+.

(**Table 2**). All these haplotypes are less frequent in semi-supercentenarians than in controls. The lowest p-value of $1.84 * 10^{-8}$ corresponds to a haplotype defined by 13 consecutive markers (chr7: 43655836–43714795). The cluster that includes this haplotype consists of 111 individual haplotypes, all carrying the same allele at the 13 consecutive markers. A p-value of $4.05 * 10^{-8}$ was detected for a second significant haplotype, located on COA1 gene and defined by eight consecutive markers (chr7: 43720429–43756081). This haplotype is in a cluster containing 112 individual haplotypes for which the same allele sequence is carried by all the individual haplotypes at the eight consecutive markers. The third significant haplotype (p-value=$4.00 * 10^{-8}$) located in COA1 gene is characterized

**Table 2.** Most significant haplotypes from each significant suggestive area emerged from the single-SNP analysis.

| Chr | Region | Haplotype | Gene | Allelic test OR | Allelic test p-value | F_105 | F_CTRL |
|-----|--------|-----------|------|-----------------|----------------------|-------|--------|
| 2 | 196515104–196992016 | GGAGCA | DNAH7 | 11.54 (3.02–65.45) | $2.34*10^{-05}$ | 0.98 | 0.82 |
| 7 | 43637796–43643835 | GTA | COA1 | 0.19 (0.1–0.36) | $4.00*10^{-08}$ | 0.35 | 0.74 |
| 7 | 43655836–43714795 | CATGATTAGTACG | COA1 | 0.18 (0.09–0.35) | $1.84*10^{-08}$ | 0.35 | 0.75 |
| 7 | 43720429–43756081 | GATGACTT | COA1 | 0.19 (0.09–0.36) | $4.05*10^{-08}$ | 0.36 | 0.75 |
| 7 | 151364264–151376555 | ACCAT | PRKAG2 | 0.13 (0.05–0.37) | $8.31*10^{-06}$ | 0.04 | 0.25 |
| 9 | 114679493–114691177 | TTATGC | UGCG | 3.23 (1.66–6.57) | $1.69*10^{-04}$ | 0.48 | 0.22 |
| 9 | 103874937–103955668 | TAA | LPPR1 | 0.13 (0.05–0.32) | $9.16*10^{-07}$ | 0.05 | 0.29 |
| 11 | 18852217–18873142 | CCTGT | . | 3.23 (1.66–6.57) | $1.69*10^{-04}$ | 0.48 | 0.22 |
| 13 | 49897577–49905581 | GCGATCG | CAB39L | 0.21 (0.09–0.44) | $1.07*10^{-05}$ | 0.10 | 0.35 |

by three consecutive markers (chr7: 43637796–43643835). It is to note that these markers are three out of the five variants identified by single-SNP analysis (i.e. rs7456688, rs10257700, rs10279856). This haplotype was found in all 109 individual haplotypes that are in the related cluster.

The most significant haplotypes of the remaining six suggestive significance areas emerging from the single-SNP analysis (located in DNAH7, PRKAG2, UGCG, LPPR1, CAB39L genes and in chr11:18852217–18873142), were reported in *Table 2*, but the haplotype cluster analysis did not identify statistically significant signals at the commonly used $5 * 10^{-8}$ significance level.

## Comparison with existing data

Next, we investigated whether the eight known 'longevity variants' reported in Erikson (*Erikson et al., 2016*) might be associated to extreme longevity in the Italian populations (*Supplementary file 7*). Only one significant SNP located in TP53 gene was identified in our study (rs1042522 nominal p-value=0.01). We also analysed the association with APOE-e4, considering the combination of rs7412 and rs429358. We showed a reduced frequency of APOE-e4 in 105+/110+ compared to controls but the difference was not significant (105+ APOE-e4 16%, CTRL_WGS APOE-e4 22%).

We then listed the SNPs identified in a previous longevity study of the Italian population (*Giuliani et al., 2018a*) and reported in *Supplementary file 8* the p-values calculated comparing 105+/110+ and CTRL (Cohort 1). Eighteen unique SNPs showed a significant p-value (nominal p-values<0.05) when considering Cohort 1. Eleven of them mapped in ESRRG (rs1436897), GUCY2EP (rs10899257), FHIT (rs2630196, rs2630173), PLCB1 (rs10485720), TBX18 (rs860844), ARID1B (rs17266366), NACAD (rs61740895, rs3735495, rs3735494), OR7G1 (rs1036224) genes. NACAD and TBX18 genes have also been associated to longevity in the Health and Retirement study, an ongoing panel survey of a nationally representative sample of men and women older than 50 years in the United States (data previously reported in *Giuliani et al., 2018b*). In *Supplementary file 8* the SNP that showed the lowest p-value in the association analysis of Cohort 1 is rs623108, located in an intergenic region downstream BLVRA gene. rs623108 is in moderate LD (r2 = 0.58 in European populations of 1000Genomes) with the SNPs reported in *Table 1*. The rs623108-A allele is correlated with rs7456688-A, highlighting the central role of this region for extreme longevity in the Italian population. A GTEx analysis showed that rs623108-A correlates with STK17A expression in heart (p-value=$7.6*10^{-13}$, NES = 0.39 for atrial appendage and p-value=$9.5*10^{-23}$ and NES = 0.36 for Left Ventricle) and in thyroid (p-value=$7.5*10^{-20}$ and NES = 0.27). eQTL violin plots are reported in *Figure 2—figure supplement 1*.

## 105+/110+ private mutations and rare variants analysis

We first performed a private mutation analysis of 105+/110+: we identified a total number of 3,446,719 private mutations exclusive of 105+/110+ (not present in CTRL) and 2,282,600 mutations showed a MAF $\leq$1% in the Genome Aggregation Database (gnomAD). *Figure 4—figure supplement 1* reports a density plot with the prevalence of private mutations for 105+/110+. Among these mutations 5055 have been predicted as damaging in more than 4 (out of 6) databases (SIFT Pred, Polyphen2 HVAR Pred, MutationTaster Pred, MutationAssessor Pred, FATHMM Pred, FATHMM MKL Coding Pred). A complete list is reported in *Supplementary file 9*. Sixty-five private mutations carried by 44 105+/110+ have also been reported in ClinVar as pathogenic or likely pathogenic.

We then considered the rare variants identified in Cohort1. First, in order to exploit the richness of our WGS data set, we considered 9,303,614 rare variants (minor allele frequency <1% in both 105+/110+ and CTRL) for association with longevity using a gene-burden association test (SKAT-O). The top hit was a region located on chromosome 17 (49,230,046 bp to 49,238,995 bp) containing NME1 gene (p-value=$6 *10^{-5}$). *Figure 4C* shows the different rare variants pattern in the group of controls and 105+/110+. A full list of all the significant genes (nominal p-value<0.01) is available in *Supplementary file 10*.

Secondly, we considered 34,852 rare and damaging mutations (predicted as damaging in more than 4 [out of 6] database) and performed a gene-burden association testing (SKAT-O). The complete list of genes identified is reported in *Supplementary file 11*. The most significant gene is PLEKHG4 (p-value=0.0011) which shows an increased burden of damaging mutations in CTRL but

not in 105+/110+. According to OMIM the protein encoded by this gene has a role in intracellular signalling and dynamics at the Golgi apparatus.

## Cardiovascular insight: analysis of somatic mutations and polygenic risk score (PRS)

The analysis of somatic mutations was performed considering those individuals whose cell count is not compromised as suggested by a recent paper (*Jaiswal et al., 2014*). For this analysis, 79 105+/110+ and 31 controls have been considered. We identified 147 somatic mutations in 80 individuals. The VAF distribution is reported in *Figure 5—figure supplement 1*.

By comparing the total number of somatic mutations between controls and 105+/110+ considering the seven analysed genes altogether (DNMT3A, TP53, ASXL1, TET2, SF3B1, PPM1D, JAK2), we found significant differences between the groups (Mann-Whitney p-value=0.00125) where 105+/110+ have a reduce burden of mutations, the median number of mutations being one for 105+/110+ and two for controls.

Moreover, Mann-Whitney U test was used to compare the number of variants in each gene between 105+/110+ and significant differences were found for DNMT3A (Mann-Whitney p-value=$9*10^{-4}$) and ASXL1 gene (Mann-Whitney p-value=0.0167).

We then calculated the prevalence of somatic mutation for each gene and observed that a general trend in which 105+/110+ have a lower prevalence (except for JAK2 gene) as reported in *Figure 5A*. Prevalence was calculated considering the number of individuals carrying at least one mutation in each gene. As expected DNMT3A and TET2 were the most commonly mutated genes in both 105+/110+ and CTRL.

Similarly to previous report (*Buscarlet et al., 2017*) the majority of somatic mutations corresponded to a C>T substitution in both 105+/110+ and controls, (*Figure 5B* and *Figure 5C*).

Next, we performed the same analysis considering those somatic mutations that are likely to disrupt the protein function (moderate and high impact variants) and that have been suggested to be drivers contributing to clonal expansion (*Genovese et al., 2014*). In this case, the total prevalence considering all seven genes was 10.13% for 105+/110+ and 0% for controls which was not statistically significant (Fisher's test p-value=0.1025). The positions identified are reported in the *Supplementary file 12*.

We, then, checked if the somatic mutations identified are listed in the COSMIC database and followed the definition of 'candidate driver somatic mutations' as reported in *Genovese et al., 2014* (*Genovese et al., 2014*) that listed in this category also the mutations reported at least seven times in hematopoietic and lymphoid malignancies. 11 mutations were identified and reported in *Supplementary file 13*.

Since the sequencing data were generated with DNA extracted from two different sources (PBMC or whole blood) a Mann-Whitney U test was performed to verify whether somatic mutations in these genes were different among cell type when considering controls and 105+/110+ separately. No statistically significant differences were found (DNMT3A: CTRL: p-value=0.2596; 105+/110+: p-value=0.3452 - TP53: CTRL: p-value=0.5000; 105+/110+: p-value=0.2417 - ASXL1: CTRL: p-value=0.2858; 105+/110+: p-value=0.1970 - TET2: CTRL: p-value=0.4565; 105+/110+: p-value=0.4744 - SF3B1: CTRL: p-value=0.1974; 105+/110+: p-value=0.4558 - PPM1D: CTRL: p-value=0.3079; 110+/110+: p-value=0.3040 - JAK2: CTRL: p-value=0.1088; 105+/110+: p-value=0.2529).

Moreover, to check the presence of cardiovascular risk variants in 105+/110+ we calculated the five most recent polygenic risk scores (PRS) for CVD in CTRL and 105+/110+ to see if 105+/110+ are characterized by lower PRS than controls. We calculated PRS according to *Natarajan et al., 2017* (67 SNVs), *Nelson et al., 2017*, *UK Biobank CardioMetabolic Consortium CHD Working Group et al., 2019* (300 SNVs), *van der Harst and Verweij, 2018* (661 SNVs), *Khera et al., 2018* (6630150 SNVs). No significant differences were found (p-value>0.05) (*Figure 5—figure supplement 2*). Standard deviation (that can be considered as proxy for heterogeneity) of PRS tends to be higher among 105+/110+ but the difference was not significant. PRS were also calculated for the top five leading causes of death with a genetic component—heart disease, cancer, stroke, Alzheimer disease, and diabetes following the list of SNPs reported in *Erikson et al., 2016*, and 105+/110+ did not show any significant difference when compared to controls (*Supplementary file 14*).

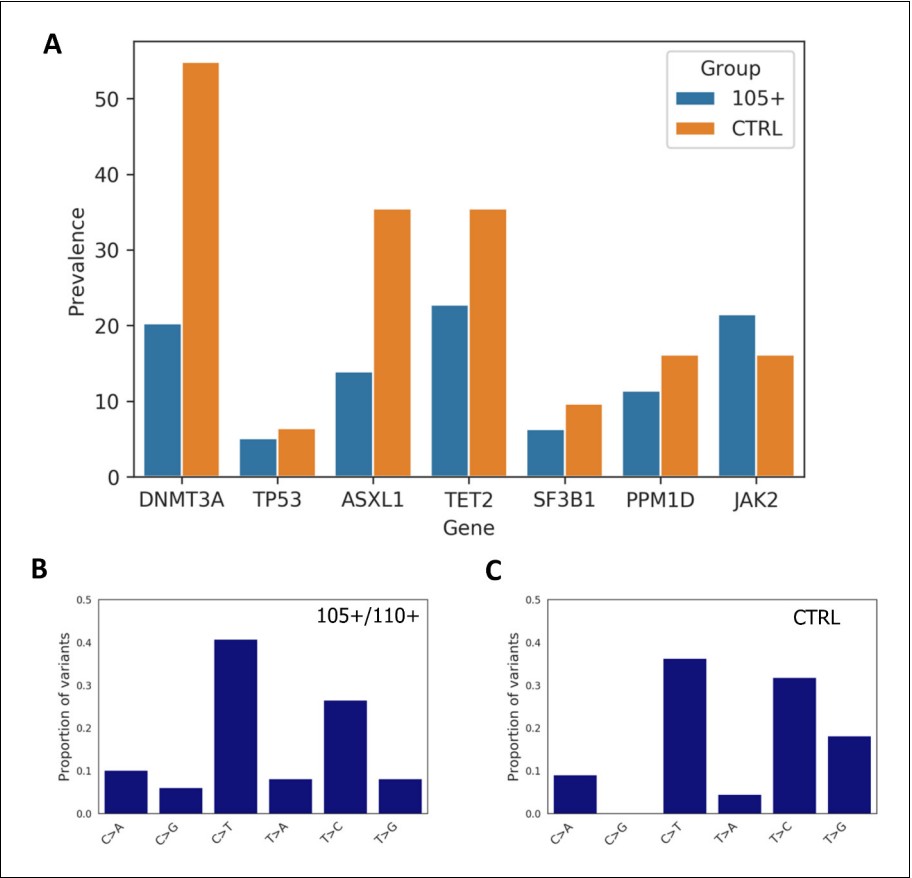

**Figure 5.** Prevalence of somatic mutations. (**A**) Prevalence of somatic mutations in 105+/110+ and controls considering the seven genes analysed. (**B,C**) the distribution of single-nucleotide substitutions types observed in 105+/110+ and CTRL.

The online version of this article includes the following figure supplement(s) for figure 5:

**Figure supplement 1.** Allelic fraction distribution of the somatic mutations observed.

**Figure supplement 2.** Boxplot with Polygenic risk scores (PRS) calculated according to different SNPs list identified from previous publication and applied to 105+/110+ and CTRL.

## Discussion

In the present study, we provide results from the first whole-genome sequencing analysis at high coverage (90X) performed in 81 105+/110+ (mean age: 106.6 ± 1.6 years) and 36 CTRL (mean age 68.0 ± 5.9 years) representative of one specific population (i.e. the Italian peninsula).

This study design attempts, for the first time, to deal with the main weaknesses encountered in the study of genetics of longevity that Sebastiani and colleagues recently highlighted:

1. A 'relaxed definition' of longevity as survival to age 85 and older, in order to increase the sample size through a meta-analysis. This inevitably increases the heterogeneity of the phenotype and to avoid this risk, we considered only individuals that reached the last decades of lifespan and individuals older than 100 years for the replication. The apparently low number of 105+/110+ is due to the fact that the recruitment of these most unique persons is complicated because of their very low number in the general population (considering individuals born in Italy in 1903, the number of people alive at age 105 was 78, given 100,000 alive at birth according to the Italian national registry ISTAT) and their delicate health conditions;

2. The issue of population heterogeneity in terms of genetic ancestry and ethnicities. This study specifically focused on one population (the Italian one) to reduce the bias due to tangled population-specific dynamics (*Giuliani et al., 2017*; *Yashin et al., 2014*), taking into account the fact that population specific evolutionary dynamics (such as demography or selection) can lead to high frequencies of certain variants linked to healthy aging or modern pathologies

(*Sazzini et al., 2016*). We selected 105+/110+ individual perfectly matched with controls for geographically origin (from North to South Italy) according to an ecological approach recently described (*Franceschi et al., 2020*; *Giuliani et al., 2018a*).

The choice of controls is challenging for studies on human longevity. Here we considered a group of healthy unrelated individuals selected from the general population as control group. We are aware that, since they are still alive, some of them may eventually become 105+/110+, but we believe that in any case this number will be very small given the low prevalence of 105+/110+ in the general population.

We identified five common variants in LD (rs7456688, rs10257700, rs10279856, rs69685881, and rs7805969) with significance at adjusted p-value 10%, all in the same region located between COA1 gene and STK17A gene. The gene-based analysis of WGS data identified STK17A gene as the most significant gene that is validated in the Cohort 2.

The U-shaped in allele frequency of rs7456688-A allele showed that these variants are peculiar of 105+/110+ individuals, and this is the first study that includes a high number of 105+/110+ to detect this signal.

All these variants were replicated in Cohort 2 (unadjusted p-values<0.05) which is made of 333 Italian centenarians (>100 years) geographically matched to 358 controls (mean age: $60.7 \pm 7.2$).

One of these five variants, rs10279856, may play a regulatory role in the region, as supported by the results obtained from risk variant inference (Riviera) and GTEx database. The SNP rs10279856 seems to play a pleiotropic role as it is an eQTL for STK17A gene and for two other genes (COA1 and BLVRA). The haplotype-based analysis confirmed that COA1 presented the most significant signal and identified a haplotype strongly associated to extreme longevity (chr7: 43720429–43756081) (p-value=$1.84*10^{-8}$). Moreover, the comparison with existing data (*Giuliani et al., 2018b*) also identified one SNP (rs623108) with potential impact on STK17A expression, indicating that different signals from different SNPs in moderate LD seems to converge on regulating the expression of COA1, STK1A, and BLVRA genes. Further functional studies are needed to elucidate the role of these genes.

Considering the four SNPs identified by Riviera analysis – that is rs10279856, rs3779059, rs849166, rs849175 – we observed that the most frequent alleles in 105+/110+ (rs10279856-G reference allele and rs3779059-A, rs849166-A, rs849175-A alternative alleles) are associated with the increase in SKT17A gene expression in heart (atrial and left ventricle), lung, nerve, and thyroid (data from GTEx portal). STK17A is involved in DNA damage response and positive regulation of apoptotic process (*Sanjo et al., 1998*) and regulation of reactive oxygen species (ROS) metabolic process. Moreover, it has been suggested that STK17A can be activated in response to external stimuli such as UV radiation and drugs (*Sanjo et al., 1998*). SNP rs7805969-A allele (located in STK17A/COA1 region) was found to be associated to systemic lupus erythematosus (SLE) in a population from Southern Brazil (*da Silva Fonseca et al., 2013*), and a reduced expression of SKT17A has been observed during the active phase of SLE disease (*Sandrin-Garcia et al., 2009*). These data suggest a possible role of this gene in DNA damage response as the variants associated to an increase of SKT17A expression (in-silico prediction) were found more frequent in 105+/110+ than controls, supporting the data by Gorbunova and colleagues on a central role of DNA repair mechanisms in aging and longevity (*Gorbunova et al., 2007*). They proposed the following sequence of events that occurs during aging: (i) mutation impairs function of genes involved in stress response and DNA repair; (2) DNA repair became more error-prone leading to accumulation of DNA damage; (3) this process accelerates age-related decline. In this model, genetic variants in STK17A may maintain DNA damage responses in 105+/110+, favoring healthy aging. On the contrary, autoimmune disease (such as SLE) are characterized by the accumulation of DNA double strand breaks possibly due to impaired repair (*Souliotis et al., 2016*) which is in line with data that described a reduced expression of SKT17A. These data on human extreme longevity support a recent study on lifespan in mammals which analyze evolutionary constraints at protein level and found DNA repair as one of the mechanisms allowing an extended lifespan across species (*Kowalczyk et al., 2020*).

Moreover, the most frequent genotypes in 105+/110+ (rs10279856-G reference allele and rs3779059-A, rs849166-A, rs849175-A alternative alleles) are not only associated to STK17A expression but also to a reduced expression of COA1 gene in adipose, artery, esophagus – mucosa, nerve – tibial and skin. COA1 gene is a component of the MITRAC complex (mitochondrial translation

regulation assembly intermediate of cytochrome c oxidase complex) that regulates cytochrome c oxidase assembly. MITRAC complexes regulate both the translation of mitochondrial-encoded components and the assembly of nuclear-encoded components imported in mitochondrion and in particular the respiratory chain complex I and IV. Our result constitute the first evidence of an association with longevity of nuclear loci mapping in a gene deeply involved in mitochondrial dynamics, supporting the hypothesis that nuclear/mitochondrial co-evolution may have a crucial role for human longevity and health (*Garagnani et al., 2014*). The same SNPs are associated with an increase in BLVRA expression in whole blood and a decrease of the expression of the same gene in artery (tibial) and esophagus (mucosa). The protein encoded by the BLVRA gene belongs to the biliverdin reductase family, members of which catalyze the conversion of biliverdin to bilirubin. Recently it has been established that a redox cycle based on BVRA activity provides physiologic cytoprotection as BVRA depletion exacerbates the formation of reactive oxygen species (ROS) and increase cell death. Interestingly, BLVRA contributes significantly to modulation of the aging process by adjusting the cellular oxidative status (*Kim et al., 2011*). Moreover, Biliverdin reductase A was previously shown to regulate the inflammatory response to endotoxin, by inhibiting Toll-like receptor 4 (TLR4) gene expression (*Wegiel et al., 2011*).

Considering the complexity of the trait under study, it has recently been proposed that even suggestive and marginally significant p-values can be highly informative in the case of longevity (*Erikson et al., 2016*; *Zeng et al., 2016*), an argument supported by Yashin and colleagues who showed that longevity also depends on several small-effect alleles (*Yashin et al., 2010*). In this context, the pathway analysis is crucial as the integration of many SNPs with modest p-values may identify biological functions and crucial pathways involved in longevity (*Johnson et al., 2015*). This analysis identified in several pathways enriched in our cohort: axon guidance, calcium signaling, glycine serine and threonine metabolism, long-term potentiation, melanogenesis, PPAR signaling and taste transduction (see Supplementary Material 1 for more details).

In this study, APOE-e4, the gene identified in a high number of studies on human longevity showed only a general trend but no significant association with longevity was found in Cohort 1. This is in line with recent data published by the GEHA Consortium (European project on the Genetics of Healthy Ageing) where APOE-e4 did not show association with longevity in the Italian population. Factors explaining this discrepancy are the variability of this haplotype across Europe, the cline that led to the low frequency in Italy (APOE-e4 is around 8% in South Italy), the peculiar gene-environment interaction experienced by certain birth cohort, and the gender effect (*Giuliani et al., 2018a*). The analysis of private mutations of 105+/110+ showed that some damaging variants and pathogenic variants are compatible with extreme longevity and healthy ageing (*Supplementary file 9*).

Rare variants analysis showed significant associations for the NME1 gene when all rare variants were considered, and for the PLEKHG4 (puratrophin-1) gene when only damaging rare variants were considered. NME-1 is the first metastasis suppressor gene discovered *Steeg et al., 1988* whose expression inhibits cell motility and metastasis in different human cancers. It regulates signalling pathways stimulated by various growth factors, including TGF-beta, platelet-derived growth factor, IGF1, lysophosphatidic acid, and serum, which suppresses metastasis (*Russell et al., 1998*). Recently, it has been demonstrated that NME1 is rapidly recruited to double-strand breaks promoting DNA repair (*Kaetzel et al., 2015*). PLEKHG4 is associated with Spinocerebellar ataxia, a neurodegenerative disease affecting cerebellar Purkinje cells. Atrophic Purkinje cells from these SCA patients have cytoplasmic aggregates containing Puratrophin-1 and the actin-binding protein Spectrin (*Ishikawa et al., 2005*). These regions seems to be preserved in 105+/110+ individuals who largely postpone age-related diseases and cancers, among other common diseases (*Ishikawa et al., 2005*).

The analysis of somatic mutations suggests that 105+/110+ individuals seem to be protected from accumulation of such mutations and we did not observe such an increase as would be expected considering their age. 105+/110+ individuals are characterized by a lower prevalence of somatic mutations in six out of the seven genes considered that is statistically significant for DNMT3A and ASXL1 genes. Focusing on somatic mutations with a potential impact on protein function the prevalence was not different from the control group.

This supports recent longitudinal data that showed that somatic mutations in *DNMT3A* and *TET2* genes previously linked to hematopoietic malignancies are common in the oldest old (*Genome of The Netherlands Consortium et al., 2016*).

These results show that 105+/110+ individuals seem spared from the age-related exponential increase of disruptive mutations, and this might have contributed in protecting from CVD (*Genovese et al., 2014*; *Jaiswal et al., 2014*; *Jaiswal and Ebert, 2019*).

However, it is to note that a depth of coverage of 90x is not the golden standard to call somatic mutations that require a coverage around 4000x as performed in recent studies (*Buscarlet et al., 2017*). A lower sequencing depth is less sensitive to detect low allele fraction variants. Other studies about somatic mutations have been performed considering exome sequencing data or whole genome sequencing with a 30x mean coverage only (*Zink et al., 2017*; *Jaiswal et al., 2014*; *Genovese et al., 2014* among others). The methodological variability (in term of coverage and part of the genome analysed) makes the comparison among existing studies difficult and not always possible.

On the contrary the existing PRS for CVD showed that 105+/110+ are not protected from the risk of CVD as the data showed no significant results when 105+/110+ were compared to controls. This can be due to three non-mutually excluding reasons: (1) PRS does not include population-specific dynamics and may not be specifically informative for the Italian population; (2) 105+/110+ have the same CVD risk variants of the general populations; (3) PRS score may include variants which effect can be neutralize by peculiar environmental factors or epistatic interactions. This result agreed with the studies that showed that centenarians and long-lived individuals are characterized by disease-associated variants frequencies similar to the general population (*Bonafè et al., 1999*, p. 53; *Beekman et al., 2010*; *Sebastiani and Perls, 2012*; *Freudenberg-Hua et al., 2014*; *Erikson et al., 2016*, *Erikson et al., 2016*). Using genetic data of 105+/110+ will be of extreme value in future studies to weight the role of certain 'risk' variants and could be used to identify new informative PRS.

Thus, the data reported here suggest that 105+/110+ escape CVD not because of genetic protection toward cardiovascular risk but because they are protect from the burden of somatic mutations (mainly disruptive) observed during ageing.

As follows we acknowledge the main limitation of this study:

1. The relaxed cut-off used in the discovery phase, that however is motivated by the crucial role of small-effect genetic variants in longevity (*Yashin et al., 2010*) and by the difficulties in the recruitment of 105+/110+ because of the rarity of the phenotype (*i.e.* extreme longevity);
2. The unbalanced case/control ratio where the case group is more than twice as large compared to the control group whose sample size is low (N = 36). However, the control group here analysed is – to date – the only representative cohort of all the Italian peninsula, including population clusters at the opposite ends of the cline of Italian variation (*Sazzini et al., 2020*). We decided to not include the TSI of the 1000 Genomes project in the control group first because their age is not known, secondly because they are not representative of all the Italian peninsula (as Tuscany is located in the Centre of Italy) and to maintain the matched with the 81 semi-supercentenarians who comes from Northern, Centre and Southern Italy.
3. The possibility that the signals here identified are peculiar of the Italian population. Gene-environment interactions are population-specific also because of the variability in environmental and cultural settings (dietary habits and lifestyle among others) and thus we cannot exclude that these results will not be generalizable. Only more data on semi-supercentenarians from other countries will clarify this point.

We selected a population of 105+/110+ perfectly matched with controls for geographical origin (from Northern to Southern Italy) to reduce bias due to genetic population variability, however a potential limitation is inextricably intertwined with this experimental design. Gene-environment interactions are population-specific also because of the variability in environmental and cultural settings (dietary habits and lifestyle among others) and thus it is likely that interactions with genetics may be different and not generalizable. Population-driven studies in which environmental and cultural data are included are desirable in this sense.

The major strengths of the present study are the following: (1) the design of this study based on the careful selection of individuals with more than 105 years old in order to focus on a peculiar phenotype that is extreme longevity; (2) the selection of 105+/110+ and controls in an homogenous population all matched for geographical origin; (3) the use of a second validation cohort of centenarians from the same population; (4) the high coverage of the sequencing that allowed somatic mutations analysis.

In conclusion, this study constitutes the first whole genome sequencing of extreme longevity at high coverage, that also allows somatic mutations analysis, in which 105+/110+ are compared with a group of healthy individual geographically matched. The results showed that 105+/110+ are characterized by a peculiar genetic background associated to efficient DNA repair mechanisms, as evidenced by both germline data and somatic mutations patterns (low/similar mutation load if compared to younger healthy controls from the general population). The model of 105+/110+ supports the recent literature that suggests a genetic signature in DNA repair mechanisms and clonal haematopoiesis are crucial players for cellular homeostasis and in cardiovascular events and that they can be the two central mechanisms that have protected 105+/110+ from age-related diseases, including CVDs.

# Materials and methods

## Cohorts description

### Cohort 1 (discovery cohort)

The cohort consists of 84 unrelated Italian healthy semi-supercentenarians and supercentenarians [105+/110+] older than 104 years (mean age 106.6 ± 1.6) recruited in North, Centre, and South of the Italian peninsula and 5 out of 84 were supercentenarians and 40 healthy unrelated controls (mean age 68.0 ± 5.9) matched for geographic origin (as also demonstrated by PCA).

After quality check 81 [105+/110+] and 36 controls were analyzed. Among the 105+/110+, 64 individuals (50 females and 14 males) were from Centre and Northern Italy, whereas 17 (13 females and 4 males) were from Southern Italy. Among the controls, 31 individuals (11 males and 20 females) were from Centre and Northern Italy, whereas 5 (2 males and 3 females) were from Southern Italy.

105+/110+ were born in a limited birth cohort range (1903–1909). The samples selected follow the criteria proposed by Sebastiani and colleagues (*Sebastiani et al., 2017*) that identified one percentile survival as the threshold to maximize the probability to identify genetic associations. Bloods drawn has been performed at the age reported. Extraction of genomic DNA from PBMCs or whole blood was performed using the AllPrep DNA/RNA/protein kit (QIAGEN, Hilden, Germany). DNA quantification was performed using the Quant-iT dsDNA Broad-Range Assay Kit (Invitrogen Life Technologies, Carlsbad, CA, USA) or by Quant-iT PicoGreen dsDNA Assay Kit (Thermo Fisher Scientific) according to manufacturer protocols.

Sequencing libraries were generated using the TruSeq DNA PCR-Free Library Preparation Kit (Illumina) using the 350 bp setting strictly following the manufacturer's recommendations. Part of samples were sequenced on HiSeq 2000 sequencers using the TruSeq PE Cluster Kit v3 and the TruSeq SBS Kit v3 for 2 × 100 cycles, the other on HiSeq X10 sequencers using HiSeq X Ten Reagent Kit v2.5 for 2 × 150 cycles (Illumina). Samples were pooled to limit technical variability due to sequencing and sequenced to a mean depth of 90X. A coverage between 75X and 100X was demonstrated to be advisable to reach a comprehensive and efficient analysis of variants (*Ferrarini et al., 2015*).

Each sample's raw reads were aligned to the GRCh37/hg19 reference genome by Isaac aligner (version 01.14.02.18), using a minimum PHRED quality score threshold of 20 from the 3'-end. The average starting amount of raw reads per sample is above 3.5 billion, corresponding to 357,165.21 Mb. After the alignment, the amount of mapped reads reached the 92.6% and the average depth of coverage resulted of ~110X for 105+/110+. For controls the average starting amount of raw reads per sample was 1,882,319,897 corresponding to 282,347.98 Mb. After the alignment, the amount of mapped reads reached the 95% and the average depth of coverage resulted of ~90.4X. Genotypes of variant and non-variant sites have been called with Isaac Variant Caller (version 1.0.7) (*Raczy et al., 2013*) with default parameters. In average, for 105+/110+ we identified 4,234,255 variants per case sample, of which 3,607,780 single nucleotide variants (SNVs) and 626,475 small Insertion/Deletion (INDELs). The Transition/Transversion (Ts/Tv) ratio resulted around 2.07, in line with the 2.0–2.1 range for genome-wide datasets (*DePristo et al., 2011*) indicating no major biases in the called variants. For controls we found 4,382,865 variants per control sample: 3,653,102 SNVs and 730,763 INDELs. The Ts/Tv ratio resulted around 2.071, as expected in accurate variant calling.

The WGS analysis was approved by the local Ethical Committee (S. Orsola Hospital - University of Bologna; Prot. n. 2006061707, amendment 08/11/2011; Fondazione IRCCS Cà Granda Ospedale

Maggiore Policlinico, Prot. n. 2035, amendment 30/11/2011; University of Calabria 9/9/2004 amendment on 24/11/2011).

## Cohort 2 (validation cohort)

We considered 333 centenarians (mean age: 100.4 ± 1.4) and a group of 358 unrelated healthy individuals controls (mean age: 60.7 ± 7.2). The control group is comparable with the one used for the discovery phase regarding, age, sex and geographical composition, as published in a previous study (*Giuliani et al., 2018b*). Genomic DNA was extracted from whole blood via a salting out modified protocol, or using QIAamp 96 DNA Blood Kit (QIAGEN, Hilden, Germany). DNA quantification was performed using the Quant-iT dsDNA Broad-Range Assay Kit (Invitrogen Life Technologies, Carlsbad, CA, USA) or by Quant-iT PicoGreen dsDNA Assay Kit (Thermo Fisher Scientific) according to manufacturer protocols. 200 ng of DNA were genotyped for 542,585 genetic markers with the CoreExomeChip v.1.1 array Illumina (San Diego, CA, USA). DNA samples of centenarians were recruited with the approval by the Ethical Committee of Sant'Orsola-Malpighi University Hospital (Bologna, Italy). A written informed consent form was obtained from all participants. Further approval for this study was also released in January 2011 by the Azienda-Ospedaliera Arcispedale Santa Maria Nuova ethics committee (Reggio Emilia) within the framework of the project 'GWAS of psoriatic arthritis in the Italian population' (in the present study we used only controls). The variants identified in Cohort 1 were replicated using imputed SNPs through Michigan server and filtering for Rsq >0.8 (*Das et al., 2016*).

## Samples quality check

To convert the gVCF files to the eVCF format, we first produced a reference VCF file containing the genomic position of all the variants called in centenarians and controls population (extracted using the gvcftools available at https://github.com/sequencing/gvcftools) plus the variants annotated in the clinvar database and in the genAge database. This reference VCF was used as input to the gvcf2evcf program to produce the eVCF files. The pipeline was described in *Ferrarini et al., 2015*. Then variants were annotated using the ANNOVAR tool (*Wang et al., 2010*). A Variant Tools (version2.7.0) project was created to import all the eVCF files. The project was exported to the TPED format compatible with PLINK (*San Lucas et al., 2012*). Quality controls determined concordance between sex by database and sex determined using genotypes, identified individuals with elevated missing data rates or outlying heterozygosity, identified related individuals and individuals of divergent ancestry (*Anderson et al., 2010*; *Clarke et al., 2011*). Variants in tandem repeats and homopolymer regions as well as variants with a call rate less than 98% were filtered out (*Anderson et al., 2010*; *Clarke et al., 2011*; *Erikson et al., 2016*).

Principal component analysis (PCA) was performed in Cohort 1 and five samples were excluded because eigenvectors for one of the two displayed PCs exceeded ±3 standard deviations.

The PCA was performed filtering out according to MAF (0.05), excluding all the markers with different call rates between centenarians and controls and pruning the file as follows: (a) considering a window of 50 SNPs, (b) calculating LD between each pair of SNPs in the window, (b) removing one of a pair of SNPs if the LD is greater than 0.2, (c) shifting the window 5 SNPs forward and repeat the procedure. Outlier individuals were identified according to eigenvectors for one of the two displayed PCs exceeding ±3 standard deviations from the mean calculated for that sampling location and were removed from subsequent analyses.

After quality filters statistical analysis were performed on 81 105+/110+ (18 M; 63 F) and 36 controls (13 M; 23 F). For common variants, a filter for variant with MAF <5% was applied. For rare variants, an additional filter for variants > 1% was applied.

Quality controls regarding the validation cohort were previously reported (*Giuliani et al., 2018b*) and followed the pipeline described for GWAS (*Anderson et al., 2010*) for the identification and removal of DNA samples and markers that introduce bias.

## Statistical analysis

### Common variants : single variant analysis

Single Gene-based analysis was performed by using VEGAS2 tool and p-values from all annotated common variants (https://vegas2.qimrberghofer.edu.au/). Gene-based replication has been

proposed to be regarded as a new gold standard in association studies because genes are functional units consistent across human populations and, dealing with some 20,000 genes instead of thousands or millions of SNPs, limit the burden of multiple testing (*Neale and Sham, 2004*). The default parameters have been used for this analysis (+/- 0 kb outside the gene and top 100% SNPs). We declare significance when the top-ranking genes identified analysing Cohort 1 were significant (p-value<0.05) in Cohort 2.

WGS association study was performed considering common variants using a logistic model with sex as covariate using PLINK tool version 1.9. p-Values adjustment have been made by SLIDE (*Han et al., 2009*). We declare significance at adjusted p-value 10% for the discovery phase and when p-values<0.05 in Cohort 2.

To exclude sequencing errors or artifacts, the identified SNPs were technical validated performing new genotyping experiment using MALDI-TOF - Iplex protocol (Agena) and considering a subset of 53 individuals from the Cohort 1.

Adjustment for ancestry PC was not deemed necessary since cases and controls were matched for geographic origin (North, Centre and South of Italy) as confirmed by a genomic inflation factor close to 1.00 (1.02) and results from PCA (Supplementary Material). It has recently been suggested (*Yashin et al., 2014*) that this correction could reduce the estimates of the associations of genetic variants with longevity traits as PCA analysis may use information from SNPs subject to mortality selection. In any case, we added the p-values for the top loci performing logistic model with sex, PC1, and PC2 as covariates (reported in *Supplementary file 1*).

*Figure 2C* reported the allele frequency of individuals of Cohort 1 (105+/110+ and CTRL) and Cohort 2 (100+ and CTRL) and of a new cohort of healthy controls < 50 years (N = 392, mean age = $39.5 \pm 7.2$) already described in *Giuliani et al., 2018b*. The data of Tuscans reported in 1000 Genomes project (N = 107) (http://grch37.ensembl.org/Homo_sapiens/Info/Index) is included. Missing data have been imputed using Michigan imputation server (https://imputationserver.sph.umich.edu/). Locus Zoom was used to plot the candidate regions of association (http://locuszoom.sph.umich.edu/).

GenAge database (http://genomics.senescence.info/genes/human.html) a database collecting the genes identified in studies on human aging has been queried.

RiVIERA (Risk Variant Inference using Epigenomic Reference Annotations) analysis for inference of driver variants was performed (*Li and Kellis, 2016*). This method uses a Bayesian joint-modelling of GWA association signals together with epigenomic annotations to estimate a posterior probability of disease for each SNP, given its association p-value and overlap with epigenomic marks. We applied such method to SNPs having association p-values less than $1*10^{-4}$ and used annotation from 450 epigenomic marks (including histone marks, DNase I hypersensitivity, transcription factor binding, and localization within exons). Epigenomic annotations were retrieved from *Pickrell, 2014*. RiVIERA-beta was used with the following parameters: nsteps = 100, max_epoch = 1e5, step = 1e-3 and burning fraction = 30%.

GTEx portal was used to investigate whether identified SNPs affect gene expression levels in different tissues (https://www.gtexportal.org/home/), NES is the slope of the linear regression of normalized expression data versus the three genotype categries using single tissue eQTL analysis, representing eQTL effect size. Pathway analysis was performed by means of i-GSEA4GWAS v2 considering KEGG, GO and BioCarta pathway and p-values for association calculated using annotated variants (only common variants were considered). The used algorithm combined the list of p-values for all the positions (MAF >0.05) according to SNP-mapped genes and then filtered the collection of pathways/gene sets to obtain a general p-value for each pathway. False discovery rate (FDR) was calculated as described in *Zhang et al., 2010*.

## Common variants: haplotype-based analysis

In order to perform the haplotype analysis of the suggestive significance areas emerged from the single-SNP analysis, we used Beagle version 5.1 (*Browning et al., 2018*; *Browning, 2006*; *Browning and Browning, 2007b*) to phase genotypes. Then we used Beagle 3.3.2 (*Browning and Browning, 2007b*) with the default settings to cluster the individual haplotypes at each SNP position using a localized haplotype cluster model (*Browning and Browning, 2007a*). Briefly, this model localizes each cluster to a specific marker locus $i$, and clusters together the haplotypes which have

similar probability distribution for their alleles at markers >i. For each haplotype cluster at a given locus, we defined a diallelic pseudomarker variable that is 2 if an individual haplotype is in the cluster, and 1 if it is not in the cluster. Such a definition of pseudomarker enables us to test the association between the haplotype cluster and a trait status (105+/110+ vs CTRL) in the same way as the other diallelic markers, that is using a Fisher's exact test for 2x2 tables. We used the *cluster2haps.jar* (see Online Resources) program to identify the allele sequences that define a haplotype cluster, and the *pseudomarker.jar* (see Online Resources) program to convert the haplotype clusters to diallelic markers. The R software (*R Development Core Team, 2020*) was used to perform the Fisher's exact test and to compute the Odds Ratio (OR) of the most significant haplotypes.

## Comparison with existing data

The analysis of existing data has been performed considering the SNPs previously identified in the Italian populations from GWAS data (*Giuliani et al., 2018b*).The most significant SNPs and the SNPs identified in Class A, Class B, Class E, and Class F has been merged with the results of association test performed in Cohort 1 and the significant loci were reported in *Supplementary file 8*. The different classes in *Giuliani et al., 2018a* includes different set of SNPs whose allele frequencies change according to the birth cohort (and thus age) of the group. Class A includes SNPs for which CTRL >50 years old showed higher allele frequencies than CTRL <50 years and centenarians, while similar allele frequencies were observed in centenarians and CTRL <50 years old. Class B: SNPs for which CTRL >50 years old showed lower allele frequencies than CTRL <50 years old and centenarians, while similar allele frequencies in centenarians and CTRL <50 years old were observed. Class E: these variants significantly decreased in frequency in centenarians, while CTRL <50 years old and CTRL >50 years old showed similar frequencies. Class F: these variants significantly increased in frequency in centenarians, while CTRL <50 years old and CTRL >50 years old showed similar frequencies. We selected these classes as they are more homogenous with the age of CTRL included in Cohort 1.

## 105+/110+ private mutations and rare variants analysis

Private mutations in 105+/110+ has been identified selecting those mutations that are present only in one 105+/110+ and not in any other controls (total number 3,446,719). Rare coding variants (minor allele frequency <1%) were analysed using the Omnibus Sequence Kernel Association Tests (SKAT-O) method as implemented in the *SKAT* R package. We created a null model for dichotomous trait adding sex as covariate and default settings. Damaging variants have been identified considering the positions defined 'damaging' in more than four out of six databases (SIFT Pred, Polyphen2 HVAR Pred, MutationTaster Pred, MutationAssessor Pred, FATHMM Pred, FATHMM MKL Coding Pred included in dbNSFP functional prediction version 3.0).

In Cohort 2, imputation was performed for chromosome 7 (filtering out SNPs with Rsq <0.8) using Michigan imputation server (https://imputationserver.sph.umich.edu/) and only the SNPs identified in Cohort 1 were selected. On these variants, association analysis was performed using a logistic model with sex as covariate (PLINK tool version 1.9). In this cohort, multiple test correction has been performed by Benjamin Hochberg method.

## Analysis of somatic mutations

We looked for somatic mutations according to the most important studies done so far (*Genovese et al., 2014*; *Jaiswal et al., 2014*; *Xie et al., 2014*; *Zink et al., 2017*). Briefly the definition of PUTATIVE SOMATIC MUTATIONS includes those mutations that followed the following criteria: (a) SNPs; (b) Observed once or twice in the cohort; (c) AF above 10%; (d) Failed the hypothesis that the alternate allelic count was distributed as a binomial process with mean 50% with a designed false positive rate of $10^{-5}$. We fixed the mean 50% for the binomial test as we do not have the same problems of exome sequencing in the experimental procedure described by Genovese and colleagues. We selected seven genes which are the most frequently associated to clonal hematopoiesis (ASXL1, DNMT3A, JAK2, PPM1D, TET2, TP53, SF3B1). Variants have been annotated with SnpEff [15], which predicts the effects of genetic variants to the final encoded protein, in order to detect mutations that would lead to a reduced or abnormal function of the protein. In this scenario, we identified different types of mutations with low, modifier, moderate and high impact on the protein

function. Moderate (e.g. missense variant, inframe deletion) and high impact variants (e.g. stop gained, frameshift variant) are highly likely to disrupt the protein function. Low impact variants (e.g. synonymous variant) and modifier (usually non-coding variants or variants affecting non-coding genes) have usually no impact on function or there is no evidence of impact, thus in these cases predictions are difficult. After excluding variants with alternative frequency equal to 1 or 0, we transformed the coverage distribution of centenarians so that their cumulative distribution was the same as the one of controls. We called that normalized coverage. The alternative (alt) and reference (ref) counts were recomputed as (frequency * normalized coverage). We excluded variants with normalized coverage <10 reads, with normalized alternative counts < 3 reads (*Jaiswal et al., 2014*), and that were observed in more than one control or in more than two centenarians minor allele frequency less than 0.025% (*Genovese et al., 2014*). To exclude germinal (inherited) variants [Genovese et al.], we defined as somatic mutations SNPs that failed the hypothesis that the alternate allelic count was distributed as a binomial process with mean 47.13% for control subjects and 46.43% for centenarians, indels were excluded from this analysis. The means of the binomial processes were computed as the median frequency of variants with frequency between 0.2 and 0.8. Analogously, we excluded homozygous germinal variants by testing the hypothesis that the alternate allelic count was distributed as a binomial process with mean 1% or 99%. In both cases, a significance threshold of 0.05 was used to reject the null hypothesis. Mann-Whitney U test was used to compare the average number of variants in each gene between the two groups and Fisher's exact test was used to compare the proportion of subjects with at least one mutation in the two groups. All calculations were performed in python 3.6.

## Polygenic risk score (PRS)

Initially the exploration of available genetic risk scores (GRS) for coronary artery disease (CAD) was performed, submitting the respective query to PubMed. We selected five CAD genetic risk scores that are summarized in *Supplementary file 15* and for which the data on: score algorithm, involved SNPs and the respective effect sizes used in score calculation - was extracted. For the scores 1–3, the effect sizes corresponding to the list of SNPs were retrieved from the original publications. References of scores 4–5 came from the work of Girelli and colleagues (*Girelli et al., 2017*) summarizing up-to-date association studies on CAD and identifying elsewhere recognized risk variants. As recommended by the authors of the review, EBI GWAS catalog available at http://www.ebi.ac.uk/gwas was used to extract the data. Thus, the total list of 928 (709 unique) SNPs coming from 22 GWAS studies describing variants associated with CAD was obtained after submitting the query 'coronary artery disease' to online repository. Eventually, three of the references used the detected genomic positions to construct separate CAD risk scores and they were included in our work (scores 4–6). Other 19 studies – solely claiming the newly discovered correlations with the cardiovascular disorder and not defining GRS – were omitted. The lists of SNPs involved in each of GRS are provided in Supplementary File (*Supplementary file 16*, *17*, *18*, *19*), except for score 2 (*Khera et al., 2018*) which is available at Cardiovascular Disease Knowledge Portal at http://www.broadcvdi.org/informational/data.

In general, the GRS calculation followed presented equation:

$$GRS = (A_1 * \beta_1) + (A_2 * \beta_2) + ... + (A_n * \beta_n)$$

where A stands for the genotype (wild type = 0, heterozygous = 1, homozygous = 2) and β for the effect size (expressed for example as OR or beta value) of a particular risk allele n. The.

For each participant of Cohort 1 (discovery cohort) six separate GRS were calculated using –score function within PLINK toolset. For polygenic score 1, due to a high ratio of absent risk variants among analyzed samples (above 10%), missing positions were searched for proxy SNPs using LDproxy web tool within LDlink suite available at https://ldlink.nci.nih.gov/. We performed each interrogation searching for specific RS numbers and the proxy variants were determined according to the following criteria: (1) presence in Tuscany population; (2) present in cohort I; (3) display the highest R2 (necessarily above 0.8). In the effect, 24 absent leading SNPs were successfully substituted and identified proxies were incorporated in the risk score calculation with original loadings. This operation was not considered necessary for other GRS since the ratio of absent SNPs was neglectable.

We checked whether the genetic risk score values were significantly different between centenarians and the controls: if the assumptions of sample independency, normal distribution and homogeneity of variances were met we applied Student's t-test, otherwise - Wilcoxon rank test was used. Further, since it was proposed that aging may be characterized by increase in variation of biological system – observing that with aging individuals tend to drift from the population average, we additionally checked with Levenne's test whether phenotypic groups differ in GRS variance.

Genetic risk score for each individual was calculated also considering 353 disease variants described in *Erikson et al., 2016*: 15 SNPs associated to Alzheimer disease, 83 SNPs to breast cancer, 138 SNPs to coronary disease, 25 SNPs to stroke, 65 SNPs to type 2 diabetes, 8 SNPs to colon cancer, 4 SNPs to lung cancer, 12 SNPs to pancreatic cancer and 3 SNPs to prostate cancer. These SNPs were selected from a previously described list of SNPs (*Erikson et al., 2016*) in order to generate comparable results. Logistic regression was performed to compare score distribution between cases and controls.

## Acknowledgements

This study was supported by Nestlé Research, Société des Produits Nestlé SA and by the European Union's Seventh Framework Programme to C Franceschi (grant number 602757, HUMAN); by the European Union's H2020 Project to C Franceschi and P Garagnani (grant number 634821, PROPAG-AGING); by JPco-fuND to C Franceschi (ADAGE) and by a grant of the Ministry of Science and Higher Education of Russian Federation (agreement No. 075-15-2020-808). We thank Leon Pietro Menicanti for his support on ethical issues.

## Additional information

### Competing interests

Julien Marquis, Armand Valsesia, Jerome Carayol, Frederic Raymond, Sebastiano Collino, Patrick Descombes: is affiliated with Nestlé Research, Société des Produits Nestlé SA. The author has no other competing interests to declare. Maria Giulia Bacalini: s affiliated with Nestlé Research, Société des Produits Nestlé SA. The author has no other competing interests to declare. Alberto Ferrarini: is affiliated with Menarini Silicon Biosystems SpA. The author has no other competing interests to declare. The other authors declare that no competing interests exist.

### Funding

| Funder | Grant reference number | Author |
|---|---|---|
| European Commission | 634821 | Paolo Garagnani |
| Ministry of Education and Science of the Russian Federation | 074-02-2018-330 | Claudio Franceschi |
| Nestlé Health Science | | Patrick Descombes |
| European Union 7th Framework Programme | 602757 | Claudio Franceschi |

The funders had no role in study design, data collection and interpretation, or the decision to submit the work for publication.

### Author contributions

Paolo Garagnani, Conceptualization, Supervision, Funding acquisition, Project administration, Writing - review and editing; Julien Marquis, Data curation, Software, Formal analysis, Methodology, Writing - review and editing; Massimo Delledonne, Conceptualization, Data curation, Software, Methodology; Chiara Pirazzini, Validation, Methodology, Experimental procedures; Elena Marasco, Visualization, Methodology, Writing - review and editing, Experimental procedures; Katarzyna Malgorzata Kwiatkowska, Formal analysis, Methodology, Writing - review and editing; Vincenzo Iannuzzi, Software, Formal analysis; Maria Giulia Bacalini, Resources, Investigation, Writing - review and

editing; Armand Valsesia, Resources, Data curation, Software, Methodology, Writing - review and editing; Jerome Carayol, Data curation, Software, Formal analysis, Supervision, Writing - review and editing; Frederic Raymond, Formal analysis, Writing - review and editing; Alberto Ferrarini, Data curation, Formal analysis; Luciano Xumerle, Claudia Sala, Data curation, Formal analysis, Writing - review and editing; Sebastiano Collino, Domenico Girelli, Oliviero Olivieri, Conceptualization; Daniela Mari, Beatrice Arosio, Daniela Monti, Writing - review and editing, Samples recruitment; Martina Casati, Evelyn Ferri, Benedetta Nacmias, Donata Luiselli, Davide Pettener, Francesco De Rango, Patrizia D'Aquila, Samples recruitment; Sandro Sorbi, Resources, Samples recruitment; Gastone Castellani, Software, Methodology; Giuseppe Passarino, Investigation, Samples recruitment; Luca Bertamini, Investigation, Methodology; Nicola Martinelli, Supervision, Methodology; Cristina Giuliani, Conceptualization, Resources, Software, Formal analysis, Supervision, Investigation, Methodology, Writing - original draft, Project administration, Writing - review and editing; Patrick Descombes, Conceptualization, Resources, Data curation, Software, Validation, Investigation, Methodology, Project administration, Writing - review and editing; Claudio Franceschi, Conceptualization, Resources, Funding acquisition, Writing - review and editing

### Author ORCIDs
Paolo Garagnani https://orcid.org/0000-0002-4161-3626
Katarzyna Malgorzata Kwiatkowska http://orcid.org/0000-0002-1536-5024
Cristina Giuliani https://orcid.org/0000-0002-5318-6502

### Ethics

Human subjects: The WGS analysis was approved by the local Ethical Committee (S. Orsola Hospital - University of Bologna; Prot. n. 2006061707, amendment 08/11/2011; Fondazione IRCCS Cà Granda Ospedale Maggiore Policlinico, Prot. n. 2035, amendment 30/11/2011; University of Calabria 9/9/2004 amendment on 24/11/2011). DNA samples of centenarians were recruited with the approval by the Ethical Committee of Sant'Orsola-Malpighi University Hospital (Bologna, Italy). A written informed consent form was obtained from all participants. Further approval for this study was also released in January 2011 by the Azienda-Ospedaliera Arcispedale Santa Maria Nuova ethics committee (Reggio Emilia) within the framework of the project "GWAS of psoriatic arthritis in the Italian population" (in the present study we used only controls). Informed consent, and consent to publish, was obtained for all the cohorts included in the study. An in depth description is reported in the Experimental procedures.

### Decision letter and Author response
Decision letter https://doi.org/10.7554/eLife.57849.sa1
Author response https://doi.org/10.7554/eLife.57849.sa2

## Additional files

### Supplementary files

• Supplementary file 1. Position identified in the comparison between 105+/110+ and CTRL with unadjusted p-values<$10^{-4}$ (logistic regression adding sex as covariate). In red are indicated the independent SNPs pruned for LD. In the last column, the p-values of the same analysis performed including PC1 and PC2 as covariates is reported.

• Supplementary file 2. Gene based analysis for common variants using VEGAS (genes with a nominal pvalue <0.01 were reported). Gene name and p-values were reported.

• Supplementary file 3. GTEx analysis for the 4 SNPs rs10279856, rs3779059, rs849166, rs849175 with credible score >0 in the Riviera analysis.

• Supplementary file 4. Significant (FDR < 0.05) KEGG pathways involved in longevity identified by iGSEA4GWAS software. The analysis has been performed considering all the annotated common variants in Cohort 1.

• Supplementary file 5. Significant (FDR < 0.05) GO pathways involved in longevity identified by iGSEA4GWAS software. The analysis has been performed considering all the annotated common variants.

• Supplementary file 6. Significant (FDR < 0.05) BioCarta pathways involved in longevity identified by iGSEA4GWAS software. The analysis has been performed considering all the annotated common variants.

• Supplementary file 7. Comparison of allele frequency for a subset of known longevity variants. The allele associated with a longer lifespan is reported as 'Protective allele'. Chromosome, position (GRCH 37/hg19), rs ID, gene name, protective allele, frequency in semi-supercentenarians, frequency in controls and nominal p-values were reported.

• Supplementary file 8. Comparison between the SNPs described in *Giuliani et al., 2018b* to the present study. The table reported six columns with the description of the SNP, chromosome, position in hg19, minor allele, gene name and the trend of allele frequency in different age groups as described in *Giuliani et al., 2018b* (Class A, B, E, and F, see legend). From columns 7–10 allele frequencies in Cohort 2 is reported as published in *Giuliani et al., 2018b*. The p-value of the association test performed between 105+/110+ and CTRL (Cohort 1) is reported in the last column.

• Supplementary file 9. List of 5055 105+/110+ private mutations predicted as damaging in more than 4 (out of 6) database (SIFT Pred, Polyphen2 HVAR Pred, MutationTaster Pred, MutationAssessor Pred, FATHMM Pred, FATHMM MKL Coding Pred).

• Supplementary file 10. Genes identified using SKAT-O method in 105+/110+ and CTRL including all rare variants (genes with a nominal pvalue <0.01 were reported). Gene name, pvalues and the number of variants is reported.

• Supplementary file 11. Genes identified using SKAT-O method in 105+/110+ and CTRL including only rare damaging variants (genes with a nominal p-value<0.01 were reported). Genes, p-value and the number of variants is reported.

• Supplementary file 12. Disruptive mutations (moderate and high impact). Genomic position (hg19), gene name, and group in which the mutation has been identified are reported.

• Supplementary file 13. List of somatic mutations identified that are reported at least seven times in hematopoietic and lymphoid malignancies using the catalogue COSMIC.

• Supplementary file 14. Logistic regression calculated considering genetic risk score for each individual for Alzheimer diseases, cancer (breast, colon, lung, pancreatic, prostate), coronary disease, stroke, and type two diabetes according to *Erikson et al., 2016*.

• Supplementary file 15. CAD genetic risk scores analyzed in Cohort 1.

• Supplementary file 16. SNPs used for PRS in *UK Biobank CardioMetabolic Consortium CHD Working Group et al., 2019*.

• Supplementary file 17. SNPs used for PRS in *Natarajan et al., 2017*.

• Supplementary file 18. SNPs used for PRS in *van der Harst and Verweij, 2018*.

• Supplementary file 19. SNPs used for PRS in *Nelson et al., 2017*.

• Transparent reporting form

## Data availability

RAW sequencing data cannot be released on a public database owing to the peculiar age of the probands (in the Italian population, semi-supercentenarians, people who are 105+ years old, occur at a prevalence rate of about 1 per 69,243, and supercentenarians, people who are 110+ years old, occur at a rate of 1 per 3,576,212, according to the Italian National Institute of Statistics (ISTAT 2015)). Interested researchers would need to apply with a research proposal to gain access to the raw data. This would be evaluated by an institutional review board composed of prof. Paolo Garagnani, prof. Stefano Salvioli and prof. Giuseppe Passarino, who are responsible for the ethical permission and for the data. Requests should be sent to stefano.salvioli@unibo.it. Aggregated data for controls and semi-supercentenarians is provided on figshare (Dataset title "Whole-genome

sequencing of semi-supercentenarians"). The allele frequencies of 5,5 MLN variants in Italian semisupercentenarians and controls are also available on figshare.

The following dataset was generated:

| Author(s) | Year | Dataset title | Dataset URL | Database and Identifier |
|---|---|---|---|---|
| Garagnani P, Giuliani C, Franceschi C | 2021 | Whole-genome sequencing of semi-supercentenarians | https://doi.org/10.6084/m9.figshare.12367085.v1 | figshare, 10.6084/m9.figshare.12367085.v1 |

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
