## [Decision Letter]

**Acceptance summary:**

The manuscript provides the genetic landscapes of the largest to date of semi- and super-centenarians by whole genome sequencing analysis and identifies a unique signature implicating an improved genome maintenance in longevity in humans. Given the rareness of extremely long lived people in the population, the results will certainly be of interest to the community of researchers studying the biology of aging, albeit exploratory worthy of further investigation.

**Decision letter after peer review:**

Thank you for submitting your article "Whole-genome sequencing of semi-supercentenarians identify DNA repair and cardiovascular disease protection signature" for consideration by *eLife*. Your article has been reviewed by 3 peer reviewers, and the evaluation has been overseen by a Reviewing Editor and Jessica Tyler as the Senior Editor. The following individual involved in review of your submission has agreed to reveal their identity: Alan R Shuldiner (Reviewer #1).

The reviewers have discussed the reviews with one another and the Reviewing Editor has drafted this decision to help you prepare a revised submission.

Summary:

This manuscript by Garagnani and colleagues reports a whole genome sequencing (WGS) study of a unique cohort of 81 semi/supercentenarians and 36 controls from Italy and a replication study involving geographically matched 333 centenarians and 358 controls. The authors find suggestive evidence for association of a common variant near STK17A with longevity from the WGS analysis and provide evidence for replication based on imputed genome-wide chip data. By performing rare variant analysis, the authors implicate a role of DNA damage response/repair in longevity. The authors also present data showing that semi/super-centenarians have fewer clonal hematopoiesis of indeterminate potential (CHIP) variants than would be expected, suggesting better DNA integrity/repair in the oldest old. The study represents the largest whole genome sequence study of a rare population of exceptionally extreme longevity involving semi/supercentenarians and will certainly be of interest to the community of researchers studying the biology of aging. All three reviewers agree that this study is unique and should be of broad interest. However, as detailed below, many concerns were also raised, relating in large part to insufficient power of detecting longevity-associated rare variants, inadequate clarity in the authors' methods, the limitations of their approach, and data presentation/interpretation in the current version.

Essential revisions:

1. The SNP rs7456688 that is included among the top association in the common variant analysis is very common in Europeans (MAF = 0.36 from dbSNP). The authors report MAF=22% in controls and MAF=56% in centenarians. These numbers are way off from what would be expected. If centenarians were so enriched of the uncommon allele, the association would be detected in other GWAS of extreme human longevity. In addition, such difference in MAF from other reference panel would point to an issue of population stratification. Therefore, text, data analysis, data presentation, and discussion should be clarified to address the concerns that: (1) there is no flip of alleles in the analysis; (2) there are no artifacts in the WGS data by including a display of the pile-up of the reads; (3) the analysis is adjusted for the genome-wide genotype PC; and (4) the analysis should be replicated in other studies.

2. The main strength of this study is also the main weakness: while sample size is the largest to date of supercentenarians, it is still small for discovery of moderate/modest effect sizes. Furthermore, there is the unbalanced case/control ratio where the case group is more than twice as large compared to the control group. In this respect, detailed answers to the following key questions must be available in the present manuscript:

i. It needs to be discussed (a) why so few controls were included; and (b) how such difference in such a small group may affect the results. In addition, the authors should (c) consider the possibility of boosting power for modest case numbers by increasing the number of controls to 2-3X (or more) the number of cases; or (d) discuss why such analysis with increased control sample size could not be done.

ii. Were any cases related to one another? If so, how was this accounted for?

iii. Gene-based analyses using VEGAS2 uncovered 179 genes with evidence for association with longevity at p<0.01. Assuming 20,000-25,000 genes were queried genome-wide, we would have expected 200-250 "significantly" associated genes. Thus 179 genes is no more (in fact is less) than one would expect by chance. Further, if 179 genes were advanced to the replication phase and evidence for replication was set at p<0.05, we would expect 9 genes to replicate by chance, whereas there was just 1, again, no more (in fact less) than would be expected by chance. The statistical robustness of the STK17A association needs to be convinced.

iv. Given that the significant differences of the allele frequency between groups have not reached significance threshold, even not a suggestive one (which is 5*10-6), the authors wisely used SLIDE (Han et al., 2009); however, it seems that this adjustment doesn't work because:

a. It was designed for much larger sample size (in fact Asif et al., Mol. Psychiatry 2020, have shown that using SLIDE in small sample size may not be reliable);

b. The candidate gene association did not reach significance threshold (7x10-2=0.07 in the p adjusted SLIDE column in table 1 is bigger than 0.05); and

c. SLIDE assumes normal distribution of variants across the sliding windows especially in unbalanced case-control analyses. It needs to be clarified if normal distribution (within the sliding windows) was observed and, if not given the small sample size, how significance of the results and the resulting interpretation were affected by this bias. Consider haplotype analysis of the suggestive significance area.

v. The analysis of the rare variants is unfortunately underwhelming, but the authors could expand this part by analyzing the burden of rare and damaging variants in these sequences, and by examining the private mutations that are carried by these 81 sequences.

vi. Most of the results are focused on the association between a group of common variants and longevity, which does not really leverage the use of WGS analysis. We are told very little about the sequences of these 81 people. The authors need to include analysis that leverage the whole genome sequence data.

vii. Given the rareness of semi- and supercentenarians, the results should be presented as highly exploratory worthy of further investigation. Consider changing the title to "Whole-genome sequencing analysis of semi-supercentenarians".

3. The authors describe the lengths they went through to make sure that semi/super-centenarian cases and controls were ethnically matched and provide data in the form of geographic distribution and principal components. They also mention the well-known north-south cline in genetic ancestry in Italy.

i. Looking at Figure 1, it appears that there are a higher proportion of cases that originate from the south compared to controls (only 3 <10%) of the controls come from the south whereas approximately 1/3 of cases came from the South. PCs should be included in the genetic analyses to help account for cryptic population substructure/stratification.

ii. Since there is clear dichotomy between north and south it would be interesting to present the difference in allele frequency between the two regions (central Italy is negligible as there are only 3 exceptional long lived subjects without matched controls) and how it affects the overall p value.

iii. Franceschi et al. (Aging 2000) reported the significant F:M ratio differences between north and south. It would be interesting to see if it affects allele frequency.

4. Perhaps the most intriguing findings of this manuscript is that the oldest old have fewer CHIP variants than would be expected. It is essential that they provide clear and complete details in the analysis of identifying the CHIP variants.

i. The authors appeared to do an adequate job in identifying the CHIP variants they reported based on relatively deep sequencing and reasonable allelic imbalance cut-offs.. However, somatic mutation rates appear to be very high (Figure 5). Did 80.6% of 68 year old controls actually have somatic mutations in DNMT3A?

ii. It is not clear how the authors distinguish between somatic mutations and rare, germline mutations without using the AF. This analysis should not use the MAF.

iii. The authors should discuss whether a depth of coverage of 90X is sufficient to detect somatic mutations.

iv. Were the bloods drawn at the time the cases were indeed semi- and super-centenarians or possibly years before?

5. Some of the statements about the lack of difference between PRS in different populations maybe an issue of power or of PRS that are too restrictive. The authors could expand their analyses by using PRS that includes SNPs that did not meet the level of genome wide significance

6. Clarify how the enrichment analyses were conducted, since it is not clear what genes were included in these analyses.

7. In Discussion, include a statement about generalizability to populations beyond Italy where diet and lifestyle and their interactions with genetics may be different.

8. The manuscript requires edits for many grammatical errors and awkward sentence structure.

---

## [Author Response]

Essential revisions:1. The SNP rs7456688 that is included among the top association in the common variant analysis is very common in Europeans (MAF = 0.36 from dbSNP). The authors report MAF=22% in controls and MAF=56% in centenarians. These numbers are way off from what would be expected. If centenarians were so enriched of the uncommon allele, the association would be detected in other GWAS of extreme human longevity. In addition, such difference in MAF from other reference panel would point to an issue of population stratification. Therefore, text, data analysis, data presentation, and discussion should be clarified to address the concerns that: (1) there is no flip of alleles in the analysis; (2) there are no artifacts in the WGS data by including a display of the pile-up of the reads; (3) the analysis is adjusted for the genome-wide genotype PC; and (4) the analysis should be replicated in other studies.

We thank the reviewers for this comment and we carefully checked all the above-mentioned points:

1. We checked the direction and strand of the allele and there is no flip of alleles in the analysis thus the frequency reported are correct. Moreover, the signal we described come from an entire region as Reviewers can see from Supplementary file 1.

We are confident that no artefact occurred in this region as we genotyped the same SNPs with a different technology (Sequenom MassARRAY iPLEX) (in the paper referred as “Technical validation”) in 53 individuals included in Cohort 1 and all the genotypes perfectly match.

In the text we stress the following sentence: “To exclude sequencing errors or artifacts, the identified SNPs were technical validated performing new genotyping experiment using MALDI-TOF – Iplex protocol (Agena) and considering a subset of 53 individuals from the Cohort 1”

2. We reported the pile-up of the region as requested. See Author response images 1 and 2.

**Author response image 1. sa2fig1:** The identified area (STK17A) was reported for two centenarians as an example. All the other data are uniform and comparable to this picture.

**Author response image 2. sa2fig2:** The identified area (STK17A) was reported for two controls as an example. All the other data are uniform and comparable to this picture.

3. The analysis is not adjusted for the first Principal Components (PCs). This is because other papers suggested to not include the first PCs as covariate in ageing studies if process of mortality selection may have occurred in the population of controls (Yashin et al. 2014 – “Genetic Structures of Population Cohorts Change with Increasing Age: Implications for Genetic Analyses of Human aging and Life Span”). The mean age of our controls is 68.0 ± 5.9 and we do not have a WGS of young population of controls to use to calculate PCs to include as covariates as suggested in the paper. The only Italian WGS data available are the one here reported and the Tuscans in 1000 Genomes. The latter cannot be included as they are not representative of the Italian peninsula but of the Centre of Italy only.

However as requested by reviewers we run again the analysis and we include in Supplementary file 1 a new column with the nominal p-values (logistic regression) after including PC1 and PC2 as covariates.

4. The signal on the same region was described in a second cohort of Italian centenarians. We have clarified this point in the text as follows:

“The five variants (rs7456688, rs10257700, rs10279856, rs69685881, rs7805969) identified in Cohort 1 were evaluated for association with longevity in the validation Cohort 2. Cohort 2 includes 333 centenarians (mean age: 100.4 ± 1.4) and a group of 358 unrelated healthy individuals (controls; mean age: 60.7 ± 7.2), genotyped on the Illumina 550k array platform. The analysis of Cohort 2 imputed genotypes (see Materials and methods) returned nominal p-values < 0.05 for all the 5 variants, as reported in Table 1”.

These are all the genetics data produced to date for Italian centenarians and new recruitments are impossible in this period.

However, we are aware that the frequencies are different from the data from 1000 Genomes and we are aware that our control group has a reduce sample size (we discussed this point in the discussion). For this reason, we added Figure 2C with the frequency of rs7456688-A in all the cohorts analyzed and we added imputed data from another published group of individuals < 50 years (Giuliani et al. 2018).

We hypothesized that these loci have never been described before because they are peculiar of the cohorts of 105+/110+ as the new Figure 2C reported. No other studies have the possibility to analyze an “high” number of semi-supercentenarians with more than 105 years.

The allele (rs7456688-A) increases in frequency in 105+/110+ (evident by the significant p-value) and the signal is detectable only when this group is included. If we perform a logistic regression comparing the group of 105/110 with all the controls (Cohort 1 + Cohort 2 – N=394) the p-value is still significant (p-value = 5.366e-05).

Considering all these data we modified the text adding the following sentences:

“Figure 2C reports the allele frequency of rs7456688-A in individuals from Cohort 1 (105+/110+ and CTRL) and Cohort 2 (100+ and CTRL), and of another cohort of healthy controls < 50 years (N=392, mean age = 39.5± 7.2) already described in (Giuliani et al. 2018). The Tuscans data (TSI) reported in 1000 Genomes project is also included. The pattern follows a U-shaped with the highest allele frequency observed for 105+/110+, indicating the relevant role of the variant in extreme longevity.”

We added this point in the discussion:

“The U-shaped in allele frequency of rs7456688-A allele supports that these variants are peculiar of 105+/110+ individuals and this is the first study that include a high number of 105+/110+ to detect this signal.”

As requested, we also calculated the frequency of the same variants in Northern, Centre, Southern Italy and no significant differences were detected (we added this sentence “All the identified signals described in Supplementary file 1 showed a non-significant p-values in the analysis based on geography. This indicates that the Italian genetic structure does not bias the signals identified.”).

2. The main strength of this study is also the main weakness: while sample size is the largest to date of supercentenarians, it is still small for discovery of moderate/modest effect sizes. Furthermore, there is the unbalanced case/control ratio where the case group is more than twice as large compared to the control group. In this respect, detailed answers to the following key questions must be available in the present manuscript.i. It needs to be discussed (a) why so few controls were included; and (b) how such difference in such a small group may affect the results. In addition, the authors should (c) consider the possibility of boosting power for modest case numbers by increasing the number of controls to 2-3X (or more) the number of cases; or (d) discuss why such analysis with increased control sample size could not be done.

We thank reviewer for this comment. Unfortunately, we do not have the possibility to increase the number of controls in our study and we decided to discuss this limitation. We add in the discussion the following sentence:

“(2) the unbalanced case/control ratio where the case group is more than twice as large compared to the control group whose sample size is low (N=36). […] We decided to not include the TSI of the 1000 Genomes project in the control group first because their age is not known, secondly because they are not representative of the Italian peninsula (as Tuscany is located in the Centre of Italy) and to maintain the matched with the 81 semi-supercentenarians who comes from Northern, Centre and Southern Italy”.

ii. Were any cases related to one another? If so, how was this accounted for?

All the individuals are not related to each other. We add this info in the Materials and methods section.

iii. Gene-based analyses using VEGAS2 uncovered 179 genes with evidence for association with longevity at p<0.01. Assuming 20,000-25,000 genes were queried genome-wide, we would have expected 200-250 "significantly" associated genes. Thus 179 genes is no more (in fact is less) than one would expect by chance. Further, if 179 genes were advanced to the replication phase and evidence for replication was set at p<0.05, we would expect 9 genes to replicate by chance, whereas there was just 1, again, no more (in fact less) than would be expected by chance. The statistical robustness of the STK17A association needs to be convinced.

The gene-based analysis has been performed including as input the p-values taken from the analysis of common variants. The qqplot showed deflation that is consequently evident in gene-based analysis. We move the section of gene-based analysis in the results to improve quality and we first describe the analysis of common variants and then the gene-based analysis.

Following the comment of reviewers, we realized that in the first submission we described only the top signal (STK17A gene) but the genes with evidence for replication (p-value < 0.05) were 8 (STK17A, BLVRA, MYRF, DNAH7, LOC553103, PHF14, SLC22A4, TBRG4). We calculated and reported in the manuscript the combined p-values and STK17A is the most significant (p-values=2.03*10^-9^).

iv. Given that the significant differences of the allele frequency between groups have not reached significance threshold, even not a suggestive one (which is 5*10-6), the authors wisely used SLIDE (Han et al., 2009); however, it seems that this adjustment doesn't work because:a. It was designed for much larger sample size (in fact Asif et al., Mol. Psychiatry 2020, have shown that using SLIDE in small sample size may not be reliable);b. The candidate gene association did not reach significance threshold (7x10-2=0.07 in the p adjusted SLIDE column in table 1 is bigger than 0.05); andc. SLIDE assumes normal distribution of variants across the sliding windows especially in unbalanced case-control analyses. It needs to be clarified if normal distribution (within the sliding windows) was observed and, if not given the small sample size, how significance of the results and the resulting interpretation were affected by this bias. Consider haplotype analysis of the suggestive significance area.

The p-values reported after SLIDE correction is 0.07, but we stated in Materials and methods that we described the top signals “loci with significance at adjusted p-values 10% for the discovery”. Then we followed the suggestion of reviewers and we added a haplotype analysis considering the top signals reported in Table 2S (now Supplementary file 1). As suggested by reviewers we included a new section of results and Materials and methods for the haplotype analysis. Interestingly we identified in this region a haplotype in COA1 gene with a p-value of 1.84*10-8. Thus gene-based analysis and haplotype analysis reinforced the signals we identified.

v. The analysis of the rare variants is unfortunately underwhelming, but the authors could expand this part by analyzing the burden of rare and damaging variants in these sequences, and by examining the private mutations that are carried by these 81 sequences.

We thank reviewers for this comment, and we followed this suggestion, and we added all the analyses suggested in a new section called “105+/110+ private mutations and rare variants analysis.”

vi. Most of the results are focused on the association between a group of common variants and longevity, which does not really leverage the use of WGS analysis. We are told very little about the sequences of these 81 people. The authors need to include analysis that leverage the whole genome sequence data.

We thank reviewer for this observation. The analysis of rare and private variants, of haplotypes and of the somatic mutations reported in this paper represent a novelty respect to available GWAS on longevity, and here we would prefer to focus on these topics. Future works will include other analysis (like CNV, LINE, SINE, indel).

vii. Given the rareness of semi- and supercentenarians, the results should be presented as highly exploratory worthy of further investigation. Consider changing the title to "Whole-genome sequencing analysis of semi-supercentenarians".

We perfectly agree with this viewpoint and we have changed the title accordingly.

3. The authors describe the lengths they went through to make sure that semi/super-centenarian cases and controls were ethnically matched and provide data in the form of geographic distribution and principal components. They also mention the well-known north-south cline in genetic ancestry in Italy.i. Looking at Figure 1, it appears that there are a higher proportion of cases that originate from the south compared to controls (only 3 <10%) of the controls come from the south whereas approximately 1/3 of cases came from the South. PCs should be included in the genetic analyses to help account for cryptic population substructure/stratification.

The total number of controls coming from Southern Italy in the Cohort 1 is 5. Figure 1 can be misleading because 2 individuals came from Petronà (CZ), 2 individuals from Carolei (CS) and one individual from San Lucido (CS). This is why the picture reported only three dots. The exact numbers have been clarified in the Materials and methods section. As mentioned above, to exclude that geographical genetic structure could have affected the results, we performed the analysis also including the first PCs in supplementary materials and the top signals did not vary.

We do not include the correction with PCs in the main text because (1) the genomic inflation factor (based on median chi-squared) is 1,02 (we reported this info in the manuscript) and (2) because other papers suggest to not include the first PCs as covariate in ageing studies if process of mortality selection may have occurred in the population of controls (Yashin et al. 2014 – “Genetic Structures of Population Cohorts Change with Increasing Age: Implications for Genetic Analyses of Human aging and Life Span” ). The mean age of our controls is 68.0 ± 5.9 and we do not have a WGS of young population of controls to use to calculate PCs to include as covariates as suggested in the paper. The only Italian WGS sequencing data available are the one here reported and the TSI in 1000 Genomes that we cannot use as they are not representative of the Italian peninsula but of the Centre of Italy.

However as requested by reviewers we run again the analysis and we include in Supplementary file 1 a new column with the nominal p-values (logistic regression) after including PC1 and PC2 as covariates.

ii. Since there is clear dichotomy between north and south it would be interesting to present the difference in allele frequency between the two regions (central Italy is negligible as there are only 3 exceptional long lived subjects without matched controls) and how it affects the overall p value.

We followed the suggestion of the reviewer and we compared Northern 105+/110+ to Southern 105+/110+ and we add in the text the following:

“Given the presence of the well-known Italian genetic structure we decided to perform an association analysis comparing Northern 105+/110+ to Southern 105+/110+subjects. […] This indicates that the Italian genetic structure does not bias the signals identified”

Other analysis based on geography cannot be performed because of the small sample size of CTRL (N=36).

iii. Franceschi et al. (Aging 2000) reported the significant F:M ratio differences between north and south. It would be interesting to see if it affects allele frequency.

We perfectly agree with this comment, but we decided to include the same sex:ratio in 105+/110+ from Northern and Southern Italy, thus it is unlikely that sex differences impact on our results. We added the information in Cohorts description (Materials and methods).

4. Perhaps the most intriguing findings of this manuscript is that the oldest old have fewer CHIP variants than would be expected. It is essential that they provide clear and complete details in the analysis of identifying the CHIP variants.i. The authors appeared to do an adequate job in identifying the CHIP variants they reported based on relatively deep sequencing and reasonable allelic imbalance cut-offs.. However, somatic mutation rates appear to be very high (Figure 5). Did 80.6% of 68 year old controls actually have somatic mutations in DNMT3A?ii. It is not clear how the authors distinguish between somatic mutations and rare, germline mutations without using the AF. This analysis should not use the MAF.

We thank reviewer for this comment, in the first submission to exclude germinal (inherited) variants we defined as somatic mutations SNPs that failed the hypothesis that the alternate allelic count was distributed as a binomial process with mean 47.13% for control subjects and 46.43% for centenarians as suggested in the paper of Genovese and colleagues. After the comment of review, we realized that the p-value we setted in the first version was too high (p-value=0.05). Thus, we decided to set a more stringent p-value (p-value < 10-5) and we filtered allelic fraction above 10% in order to address the concerns of reviewers. We then fixed the mean 50% for the binomial test as we do not have the same problems of exome sequencing in the experimental procedure described by Genovese and colleagues. All the analysis and data have been updated.

For clarity we reported the AF distribution in Figure 5—figure supplement 1.

Even after this filter the mutations in DNMT3A are 20,25% in centenarians and 54,8% in controls. Since this data come from whole genome sequencing (and not exome sequencing) it is likely that the percentage is higher than expected and difficult to compare with published papers. We added the following sentence to address this point “The methodological variability (in term of coverage and part of the genome analyzed) make the comparison of existing data difficult and not always possible.”

iii. The authors should discuss whether a depth of coverage of 90X is sufficient to detect somatic mutations.

We followed reviewers suggestion and we added the following consideration in the text:

“However, it is to note that a depth of coverage of 90x is not the golden standard to call somatic mutations that require a coverage around 4,000x as performed in recent studies (Buscarlet et al. 2017). […] Other studies about somatic mutations have been performed considering exome sequencing data or whole genome sequencing with a 30x mean coverage only (Zink et al. 2017 ; Jaiswal et al. 2014, Genovese et al. 2014 among others).”

iv. Were the bloods drawn at the time the cases were indeed semi- and super-centenarians or possibly years before?

The blood has been collected when semi and super-centenarians reached the reported age (105 or 110) and not years before. We agree that this detail is important and we reported it in the Materials and methods.

5. Some of the statements about the lack of difference between PRS in different populations maybe an issue of power or of PRS that are too restrictive. The authors could expand their analyses by using PRS that includes SNPs that did not meet the level of genome wide significance

The selection of PRS is based on the scores already described in scientific papers and already validated in other publications and in other cohorts. We prefer to not include “new” scores changing the number of SNPs included to make the results comparable with existing literature. Further analysis and a dedicated paper is needed to identify new scores in these cohorts or to use this cohorts to create new PRSs. We added this point in the discussion.

6. Clarify how the enrichment analyses were conducted, since it is not clear what genes were included in these analyses.

We thank the reviewers for this observation. Details have been included in Materials and methods as follows: “Pathway analyses were performed by means of i-GSEA4GWAS v.2. The used algorithm combined the list of p-values for all the positions (MAF> 0.05) according to SNP-mapped genes and then filtered the collection of pathways/gene sets to obtain a general p-value for each pathway. KEGG, GO and BioCarta pathways were reported.”

7. In Discussion, include a statement about generalizability to populations beyond Italy where diet and lifestyle and their interactions with genetics may be different.

Discussion has been modified accordingly. We added the following sentence: “(3) the possibility that the signals here identified are peculiar of the Italian population. […] Only more data on semi-supercentenarians from other countries will clarify this point.”

8. The manuscript requires edits for many grammatical errors and awkward sentence structure.

Some part of the manuscript has been rephrased and check for grammatical errors.